



# Simulation, Precursor Analysis and Targeted Observation Sensitive Area Identification for Two Types of ENSO using ENSO-MC v1.0

Bin Mu[1,*], Yuehan Cui[1,*], Shijin Yuan[1], and Bo Qin[1]

[1]School of Software Engineering, Tongji University, Shanghai, China
[*]These authors contributed equally to this work.

**Correspondence:** Shijin Yuan (e-mail: yuanshijin2003@163.com)

**Abstract.** The global impact of an El Niño-Southern Oscillation (ENSO) event can differ greatly depending on whether it is an Eastern-Pacific-type (EP-type) event or a Central-Pacific-type (CP-type) event. Reliable predictions of the two types of ENSO are therefore of critical importance. Here we construct a deep neural network with multichannel structure for ENSO (named ENSO-MC) to simulate the spatial evolution of sea surface temperature (SST) anomalies for the two types of events.

We select SST, heat content, and wind stress (i.e., three key ingredients of Bjerknes feedback) to represent coupled ocean-atmosphere dynamics that underpins ENSO, achieving skillful forecasts for the spatial patterns of SST anomalies out to one year ahead. Furthermore, it is of great significance to analyze the precursors of EP-type or CP-type events and identify targeted observation sensitive area for the understanding and prediction of ENSO. Precursors analysis is to determine what type of initial perturbations will develop into EP-type or CP-type events. Sensitive area identification is to determine the regions where

initial states tend to have greatest impacts on evolution of ENSO. We use saliency map method to investigate the subsurface precursors and identify the sensitive areas of ENSO. The results show that there are pronounced signals in the equatorial subsurface before EP events, while the precursory signals of CP events are located in the North Pacific. It indicates that the subtropical precursors seem to favor the generation of the CP-type El Niño and the EP-type El Niño is more related to the tropical thermocline dynamics. And the saliency maps show that the sensitive areas of the surface and the subsurface are

located in the equatorial central Pacific and the equatorial western Pacific, respectively. The sensitivity experiments imply that additional observations in the identified sensitive areas can improve forecasting skills. Our results of precursors and sensitive areas are consistent with the previous theories of ENSO, demonstrating the potential usage and advantages of the ENSO-MC model in improving the simulation, understanding and observations of two ENSO types.

## 1 Introduction

El Niño-Southern Oscillation (ENSO) is a nearly periodically occurring climate signal in the tropical Pacific Ocean every 2-7 years and often grows up to be exceptionally strong under unstable air-sea interactions (Bjerknes, 1969; Philander, 1983), causing large global climatic anomalies and hence affecting many regions even far from the tropical area (Yu et al., 2012). Studies suggested that each ENSO event may differ in spatial structure, temporal evolution, amplitude and trigger (Capotondi et al., 2015; Timmermann et al., 2018). One view is that there may be two different types of ENSO, referred to as the Eastern-





Pacific-type (EP-type) event and the Central-Pacific-type (CP-type) event (Yu and Kao, 2007; Kao and Yu, 2009). And the differences in the details of sea surface temperature (SST) anomaly patterns between EP and CP events will lead to different remote teleconnection patterns and effects on the global climate (An et al., 2007; Ashok et al., 2007; Timmermann et al., 2018). In recent decades, with the increased occurrence of CP El Niño relative to EP El Niño, the predictability of two ENSO types has attracted widespread attentions (Lee and McPhaden, 2010). Improved forecasting and understanding of the two types of

ENSO are therefore of great importance.

Most studies on the simulations of two types of ENSO are based on the climate numerical models. Kug et al. (2010) used Geophysical Fluid Dynamics Laboratory Coupled Model (GDFL) to simulate the CP-type El Niño, which shows distinct spatial characteristics and dynamic processes from the EP-type El Niño. More comprehensively, Kug et al. used the climate models from the Coupled Model Intercomparison Project phase-3 (CMIP3) (Ham and Kug, 2012) and phase-5 (Kug et al., 2012) to

validate the fidelity in simulating the two types of events. The results showed that a few models can simulate the two types of El Niño, and most of models tend to simulate a single type. Accurate simulations and predictions of two types of ENSO are still of a great challenge, owing to the inherent uncertainty and diversity of ENSO (Chen and Cane, 2008; Trenberth and Stepaniak, 2001; Capotondi et al., 2015).

In the past two years, deep learning methods have paved a new and profound way to making accurate ENSO forecasts for

long lead times (Huang et al., 2019; Ham et al., 2019). For example, Ham et al. (2019) used convolutional neural network (CNN) together with transfer learning method to produce higher skills in predicting the Niño 3.4 index than current dynamical and statistical models at lead times of up to one and a half years. Yan et al. (2020) used the temporal convolutional network and empirical mode decomposition to predict each subcomponents of Niño3.4 index that were then reconstructed to improve the forecasting skills of total values. In addition to the Nino index forecasting that most models are currently focused on, deep

neural networks also show great potential for a wide range of application for the pattern predictions (Mu et al., 2019, 2021). Here we develop a spatiotemporal model of multichannel structure for ENSO (named ENSO-MC) to simulate the spatial diversity and evolution of SST anomalies patterns in the equatorial Pacific. The multichannel structure containing the complex ocean-atmosphere interactions is built to achieve skillful predictions of two types of ENSO one year in advance.

In addition to developing forecast models, understandings and observations of ENSO are also of great significance for

prediction improvement, which are two basic issues in the predictability of ENSO. In order to better understand the mechanism of ENSO occurrence, one approach is to explore the precursor of ENSO, which is the initial perturbation distribution that is most likely to develop into a CP event or an EP event. These precursors help us understand the dynamic process of ENSO and provide the potential to predict ENSO events and their types. In terms of observations, owing to the limited sampling frequency in time and sampling density in space of the current observation systems, especially ocean observation systems,

intensive observations are usually prioritized in sensitive areas. Such a strategy is called targeted observation method. The key issue in targeted observation is the identification of the sensitive areas. The initial conditions in these sensitive areas may be more important than those in other regions when predicting ENSO (Mu et al., 2015). Due to the high cost of observation over the ocean, it is a cost-effective method which can help reduce initial errors, thereby reducing prediction errors and improving prediction skills.





Precursor investigation and sensitive area identification based on numerical models and optimal perturbations, such as the linear singular vector (LSV) approach (Moore and Kleeman, 1996), the linear inverse modeling (LIM) approach (Newman et al., 2011; Vimont et al., 2014) and the conditional nonlinear optimal perturbation (CNOP) technique (Mu et al., 2003), have been applied extensively and produced meaningful results. For example, Capotondi and Sardeshmukh (2015) suggested that the initial subsurface conditions play an important distinguishing role in the generation of different ENSO types. And recent

research has also recognized the critical role of some climate patterns outside the tropical Pacific (Vimont et al., 2003; Ham et al., 2013; Chikamoto et al., 2015) that precede ENSO. Based on the method of finding the optimal initial perturbation, several studies have linked the precursor analysis with the targeted observation of ENSO events (Mu et al., 2014; Hu and Duan, 2016). Hu and Duan (2016) identified the western equatorial Pacific of the subsurface and the eastern equatorial Pacific of the surface as sensitive areas using CNOP method based on the Community Earth System Model (CESM). And it showed

that eliminating the initial errors in these sensitive areas can greatly improve ENSO prediction. Despite numerous attempts for precursory signals investigation and sensitive areas identification of ENSO, the results may vary owing to the complexity of the models. The research on improved observation, understanding and forecasting of ENSO has therefore always received critical attention.

    In this paper, based on the model ENSO-MC we proposed above that can simulate spatial patterns of SST anomalies, we

further analyze the subsurface precursors of different types of events and identify the sensitive areas with the help of saliency map interpretability method (Simonyan et al., 2013; Zeiler and Fergus, 2014). The obtained saliency map answers the question "which input pixels should be changed to yield a maximal increase in the considered output value with minimal change?" (Ebert-Uphoff and Hilburn, 2020). It indicates the sensitivity of the predicted results to the perturbations in each region of the input field and can be used to discover the initial perturbation distribution that would develop into an ENSO event, which

reveals the signals prior to the events captured by the ENSO-MC. Besides, the sensitive areas are the regions in which the small perturbations would have the greatest influence on the forecasts. Therefore, saliency map method can also help identify the sensitive areas. Since the original saliency map method is prone to noise, the smoothGrad method (Smilkov et al., 2017) is used here to help sharpen the saliency maps. Sensitivity experiments are then performed to verify the effectiveness of the identified sensitive area. The results are consistent with the previous understanding of ENSO mechanism. Our results suggest

that the ENSO-MC model may provide a powerful and promising method for simulating the seasonal-to-interannual variations of ENSO and analyzing the inherent predictability of ENSO.

    The remainder of the paper is structured as follows. Section 2 describes the ENSO-MC model for simulating two types of ENSO, including the multichannel spatiotemporal prediction neural network, detailed descriptions of the predictor selection and combined loss function. Section 3 provides the assessment of the ENSO pattern simulation performance based on the

ENSO-MC model. Then we analyze the precursors in heat content for two types El Niño events and La Niña events in Section 4, and identify the sensitive areas of targeted observation for ENSO in Section 5. Summary and discussion are presented in Section 6.





## 2 ENSO-MC: Simulation model for two types of ENSO

### 2.1 Multichannel spatiotemporal structure

Here we develop a spatiotemporal model of multichannel structure named ENSO-MC to generate SST pattern sequence for ENSO forecasts. As shown in Fig. 1, the ENSO-MC is constructed based on the encoder-decoder architecture (Sutskever et al., 2014), consisting of the convolutional long short-term memory (ConvLSTM) layers (Xingjian et al., 2015), convolutional/deconvolutional layers, and pooling/upsampling layers. ENSO-MC takes the grid observations of monthly SST, heat content, zonal wind and meridional wind for $T_{in}$ consecutive months with a dimension of $4 \times T_{in} \times 80 \times 160$ ($1° \times 1°$ resolution

over $40°N - 40°S, 120°E - 80°W$) as input. After multiple experiments, here we set $T_{in}$ to 12, which is the best input sequence length. In order to establish the information connection and transmission between these ocean-atmosphere data, we treat each type of input variable as a channel in ConvLSTM and thus there are four channels for the input data ($T_{in} \times 80 \times 160 \times 4$). Three ConvLSTM layers and subsequent pooling layers are used for encoding, and three ConvLSTM and upsampling layers are used for decoding to extract nonlinear air-sea interactions underpinning ENSO. The final output of the model is the SST gridded

data for the next $T_{out}$ months.

In order to preserve oceanic processes information for a long time for ENSO forecast, we add skip-layer connection and states connection between the encoder and the decoder on different spatial scales. For each ConvLSTM layer in the encoder, the feature maps of all time steps are fused into one feature map and the weight of each time step is automatically determined through the attention mechanism. These fused feature maps are attached to the corresponding ConvLSTM layers in the decoder

to achieve skip-layer connection. And in the states connection, the hidden states output by the ConvLSTM layers in the encoder are reserved for the corresponding layer when the decoder is initialized.

### 2.2 Selection of physical variables

Using deep neural networks for ENSO simulation is essentially a data-driven method (Reichstein et al., 2019) that is good at mining complicated relations hidden in multidimensional observations of the climate system. Therefore, in addition to building

a suitable network to well fit the data, it is also important to choose appropriate predictors that well represent ENSO physical processes to train the model (Reichstein et al., 2019).

First, SST is selected as one of the predictors since it is a source of ENSO predictability and a direct reflection of the occurrence of ENSO event. As the slow-evolving thermal anomaly in the subsurface ocean that provides a key long-lasting memory for ENSO prediction (e.g., Zhang and Levitus, 1997; Tang et al., 2018), we then choose heat content (vertically

averaged oceanic temperature in the upper 300 m) as the second attribute , which is closely related to the recharge-discharge oscillator point of view (Jin, 1997a, b; Meinen and McPhaden, 2000; McPhaden, 2003). Third, the westerly wind burst is believed to be an important trigger for El Niño events (Gebbie et al., 2007; Hu et al., 2014; Menkes et al., 2014; Chen et al., 2015; Fedorov et al., 2015) and the atmospheric noise from the wind can be a limit for ENSO predictability (Latif et al., 1988; Moore and Kleeman, 1999). We therefore select SST, heat content, and wind stress, in accordance with Bjerknes feedback, to

simulate the air-sea interactions responsible for ENSO.

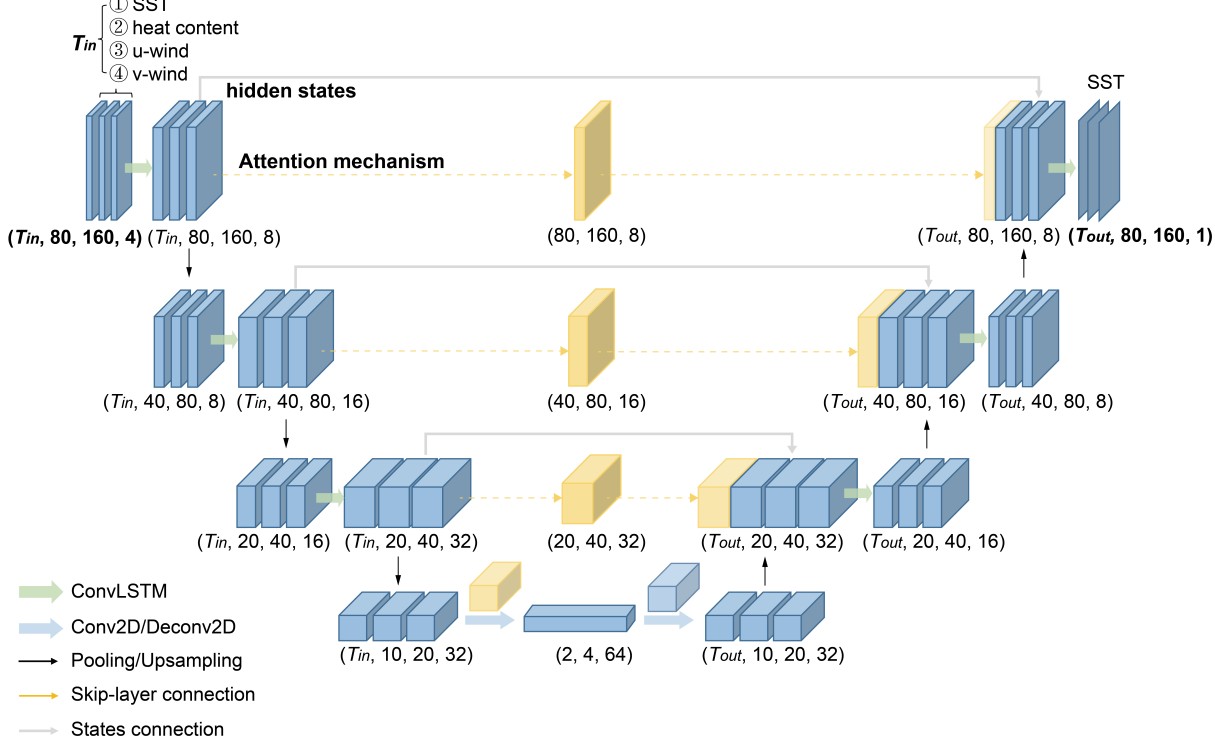

**Figure 1.** The encoder-decoder architecture of ENSO-MC for SST pattern sequence prediction. The encoder contains three ConvLSTM layers and each layer is followed by a pooling layer, and the last layer is a convolutional layer, which allows extracting spatial and temporal features. And the decoder comprises one deconvolutional layer, three ConvLSTM layers and three upsampling layer that restores the features to the same size as the initial spatial dimension (80×160). The model uses skip-connection with attention mechanism and state-connection between the encoder and the decoder to improve forecasting skills. The input variables are the SST, heat content, zonal wind and meridional wind for $T_{in}$ consecutive months over $40°N - 40°S$, $120°E - 80°W$ (80×160, $1° \times 1°$ resolution) and each type of variable is input to the model as a channel. The output of the model is SST pattern sequence for the next $T_{out}$ months.

We utilize Simple Ocean Data Assimilation (SODA) reanalysis data set consisting of sea surface temperature, heat content and wind stress gridded variables from 1871 to 2008 to train the ENSO-MC model, with the resolution $1° \times 1°$. The domain over $40°N - 40°S$, $120°E - 80°W$ is utilized. To avoid possible overfitting, the data set is divided into training set and validation set according to the ratio of 4 to 1. Then we test the performance of model using NCEP Global Ocean Data Assimilation System (GODAS) and ERA-Interim data (2010-2019), where remove the data that is already in the training set (1981-2008), and there is a one-year gap between training set and test set to reduce the possible influence of oceanic memory.





## 2.3 Loss function

In our experiments, we combine the losses based on Mean Squared Error (MSE), Structural Similarity Index (SSIM) (Wang et al., 2002) and Gradient Difference Loss (GDL) (Mathieu et al., 2015):

$$\mathcal{L} = \lambda_{mse}\mathcal{L}_{mse}(Y,\hat{Y}) + \lambda_{ssim}\mathcal{L}_{ssim}(Y,\hat{Y}) + \lambda_{gdl}\mathcal{L}_{gdl}(Y,\hat{Y}). \tag{1}$$


$Y(\hat{Y})$ denotes the observed (predicted) SST pattern sequence. $\lambda$ represents the weight of each loss. Specifically, MSE measures the discrepancy of each pixel in the sea surface temperature field:

$$\mathcal{L}_{mse}(Y,\hat{Y}) = \frac{1}{T}\sum_{t=1}^{T}||Y_t - \hat{Y}_t||^2, \tag{2}$$

where $T$ represents the length of the prediction sequence. Since the pixels on the ocean field have strong inter-dependencies
especially when they are spatially close to one another, we introduce a loss based on SSIM to measure the global structural differences. Given two fields $x$ and $y$, the structural similarity of them can be derived as follows:

$$SSIM(x,y) = \frac{(2\mu_x\mu_y + c_1)(2\sigma_{xy} + c_2)}{(\mu_x^2 + \mu_y^2 + c_1)(\sigma_x^2 + \sigma_y^2 + c_2)}, \tag{3}$$

where $\mu$ is the average of an field, $\sigma^2$ is the variance and $\sigma_{xy}$ is the covariance of the two fields. $c_1$, $c_2$ are constants used to maintain the calculations stable. The range of SSIM is from 0 to 1, and when two fields are the same, the value of SSIM is 1.
Therefore, we construct the SSIM-based loss function as

$$\mathcal{L}_{ssim}(Y,\hat{Y}) = \frac{1 - \frac{1}{T}(\sum_{t=1}^{T} SSIM(Y_t, \hat{Y}_t))}{2}. \tag{4}$$

In addition, gradient information of gridded variables is important for the model to understand changes in the sea temperature field. In order to more accurately predict the spatial location of the SST anomalies and their development in the Pacific, we use the GDL to measure the gradient difference of the surface sea temperature field:


$$\mathcal{L}_{gdl}(Y,\hat{Y}) = \frac{1}{T}\sum_{t=1}^{T}\sum_{i,j}\left||Y_{i,j}^t - Y_{i-1,j}^t| - |\hat{Y}_{i,j}^t - \hat{Y}_{i-1,j}^t|\right|^2 + \left||Y_{i,j-1}^t - Y_{i,j}^t| - |\hat{Y}_{i,j-1}^t - \hat{Y}_{i,j}^t|\right|^2, \tag{5}$$

where $i,j$ denote the pixel position on the sea surface temperature field.

## 3 Simulation of two types of ENSO

The simulation ability of ENSO-MC for different types of events is evaluated in this section. We first select the typical EP El Niño, CP El Niño and La Niña events in recent years to validate the forecast skills on individual events in detail. The forecasts
of spatial patterns and Niño 3.4 index time series are compared with observations. Besides, the correlation skills for all targeted seasons of the model are further validated.



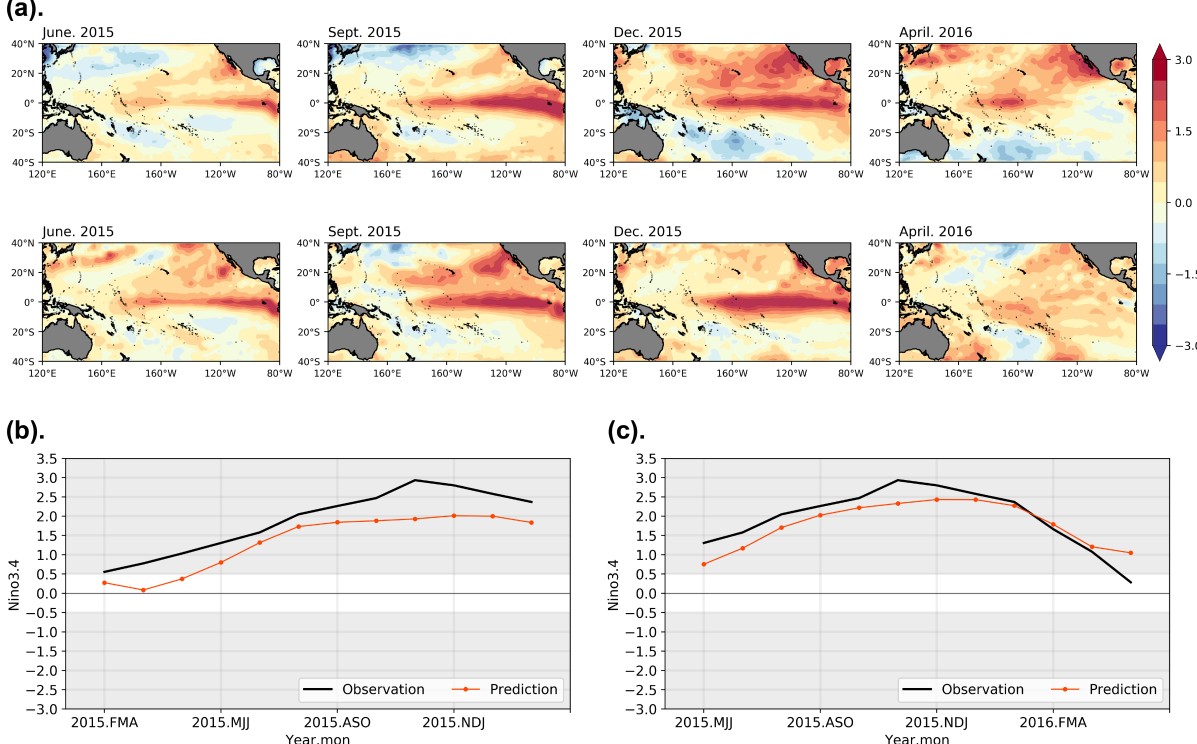

**Figure 2.** The SSTA patterns and index prediction results for 2015/16 El Niño using ENSO-MC. (a). Spatial development of SST anomalies predicted (the first row) ahead of one year compared with the real observations (the second row) for the onset, growth, mature and decay phase. (b). Nino3.4 index time series forecast initiated in the FMA season using ENSO-MC (red) and the observed Nino3.4 index (black). (c). Nino3.4 index time series forecast initiated in the MJJ season using ENSO-MC (red) and the observed Nino3.4 index (black).

We validate the performance of ENSO-MC for simulating the latest extreme El Niño event in 2015/16. It can be classified as an EP-type event, and some studies suggested that the 2015/16 El Niño appears a mixed EP and CP patterns (Santoso et al., 2017). Figure 2(a) compares the predicted spatial evolution of SST anomalies for 2015/16 event (the first row) ahead of one

year and the corresponding observed patterns (the second row). The four months shown in the Fig. 2(a) (June, September, December and April) represent the main states involved in the phases of ENSO evolution. The temperature anomalies emerge in the eastern equatorial Pacific (June. 2015), which are then amplified and spread to the central equatorial Pacific (Sept. 2015). When the event reaches the mature stage (Dec. 2015), the center of the anomalies tends to move toward the central equatorial Pacific, eventually decaying in spring 2016 (April. 2016). The prediction of SST anomalies development in the

equatorial Pacific is reasonable agreement with the observation results. But in the subtropics, there is a strong warm bias in the northeastern Pacific during the mature and decay stages. And in the South Pacific, there is more cooling in the model than observed. According to the predicted SST anomalies patterns, the time series of Nino3.4 index is further calculated (Figs. 2(b), (c)). The amplitude and temporal evolution of the 2015/16 El Niño for the one-year-lead forecast of ENSO-MC are



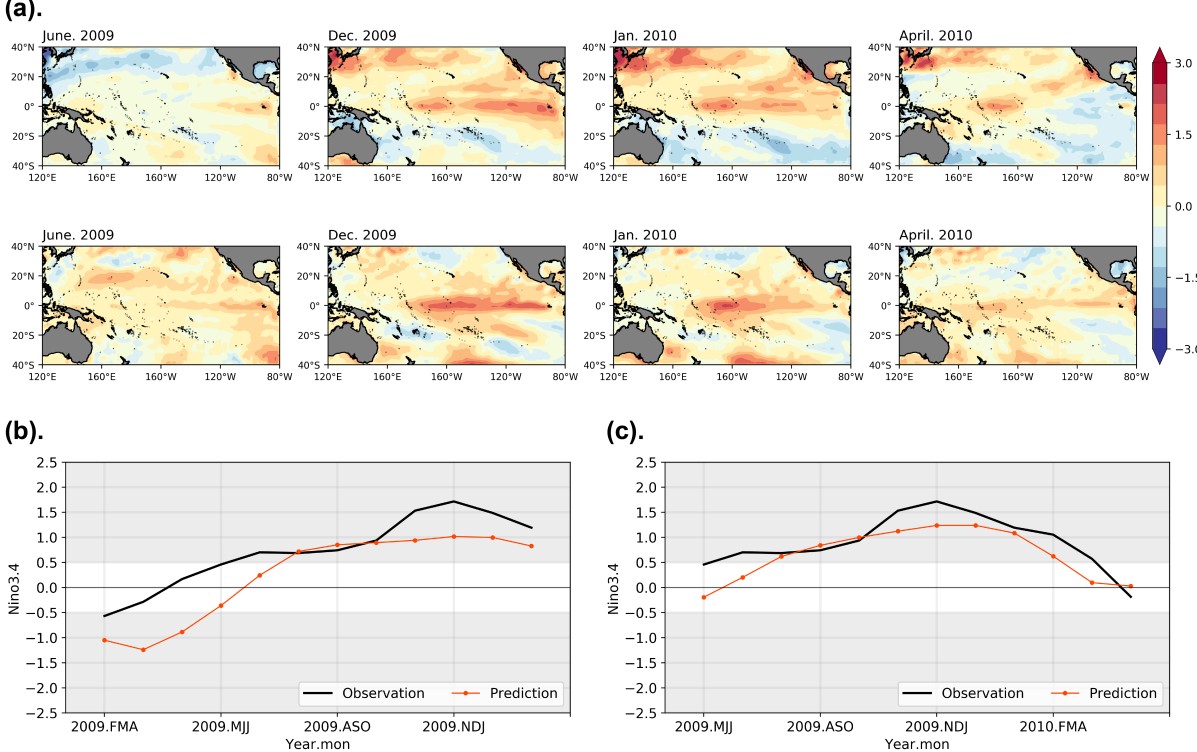

**Figure 3.** The SSTA patterns and index prediction results for 2009 El Niño using ENSO-MC. (a). Spatial development of SST anomalies predicted (the first row) ahead of one year compared with the real observations (the second row) for the onset, growth, mature and decay phase. (b). Nino3.4 index time series forecast initiated in the FMA season using ENSO-MC (red) and the observed Nino3.4 index (black). (c). Nino3.4 index time series forecast initiated in the MJJ season using ENSO-MC (red) and the observed Nino3.4 index (black).

almost consistent with observations, although with weaker amplitude compared to observed Nino3.4 index. And the forecasts
initiated during the May-June-July (MJJ) season (Fig. 2(c)) agree a bit more favorably with the observations than initiated
during February-March-April (FMA) 2015 (Fig. 2(b)).

By contrast, the prediction results of 2009 event, which is known as a CP El Niño, are shown in Fig. 3. The spatial patterns
of SST anomalies predicted (the first row) and observed (the second row) are shown in Fig. 3(a). The observations show a
pronounced central Pacific warming. Specifically, the SST anomalies appear (June. 2009), grow (Dec. 2009) and reach the
maturity (Jan. 2010) in the central Pacific, whose meridional shift on the equator is less obvious. The model captures most
of these features, although there are stronger anomalies in the eastern tropical Pacific during the growth phase than observed
and weaker amplitude in the central Pacific at mature phase. And there is also some evident bias in the midlatitudes. For the
Nino3.4 index results (Figs 3(b), (c)), the ENSO-MC model exhibits the similar trend but weaker amplitude compared with the
observed values. Especially when initiated in FMA 2009 (Fig 3(b)), the model tends to underestimate the strength during the
growth phase.



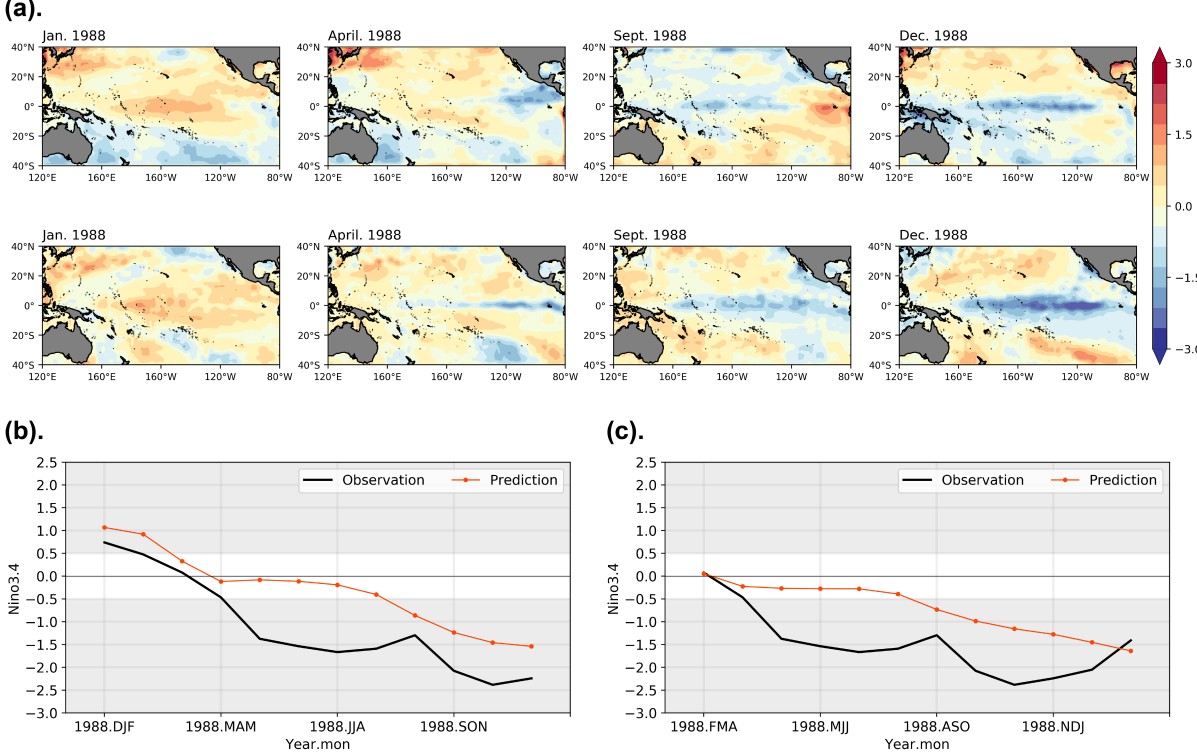

**Figure 4.** The SSTA patterns and index prediction results for 1988 La Niña using ENSO-MC. (a). Spatial development of SST anomalies predicted (the first row) ahead of one year compared with the real observations (the second row) for the transition, onset, growth, mature phase. (b). Nino3.4 index time series forecast initiated in the DJF season using ENSO-MC (red) and the observed Nino3.4 index (black). (c). Nino3.4 index time series forecast initiated in the FMA season using ENSO-MC (red) and the observed Nino3.4 index (black).

Given the different mechanisms and periods of warm El Niño and cold La Niña events, we also evaluate the simulation ability of the model for a La Nina event. Figure 4 shows the prediction results for 1988 strong La Niña. The simulated evolution of SST anomalies spatial structure along the equator are in reasonable agreement with observations (Fig. 4(a)). The cold temperature anomalies occur in the eastern Pacific, then spread to the central Pacific and reach maturity. However, the predicted cold

anomalies are weaker in amplitude, and do not extend as broad area as observed. It is evident in the Nino3.4 index time series results, where the amplitudes of the predictions are weaker than the observed values regardless of whether the initialization is performed before the event (Fig. 4(b)) or in the early stage of its development (Fig. 4(c)).

Despite some bias, the spatial and temporal evolution of SST anomalies in the tropical Pacific appear to be well simulated during the 2015/16 EP-type El Niño, 2009 CP-type El Niño and 1988 La Niña events. In addition to the evaluation of a few

typical events, the model also shows a reasonable forecast of ENSO ahead of one year for all targeted seasons (Fig. 5). Figure 5(a) shows the correlation skill of Nino3.4 index forecasts for the GODAS validation data from 1982 to 2019, which are initiated in each calendar month and predicted for a lead of up to 18 months. The correlation skill in the model is above 0.5



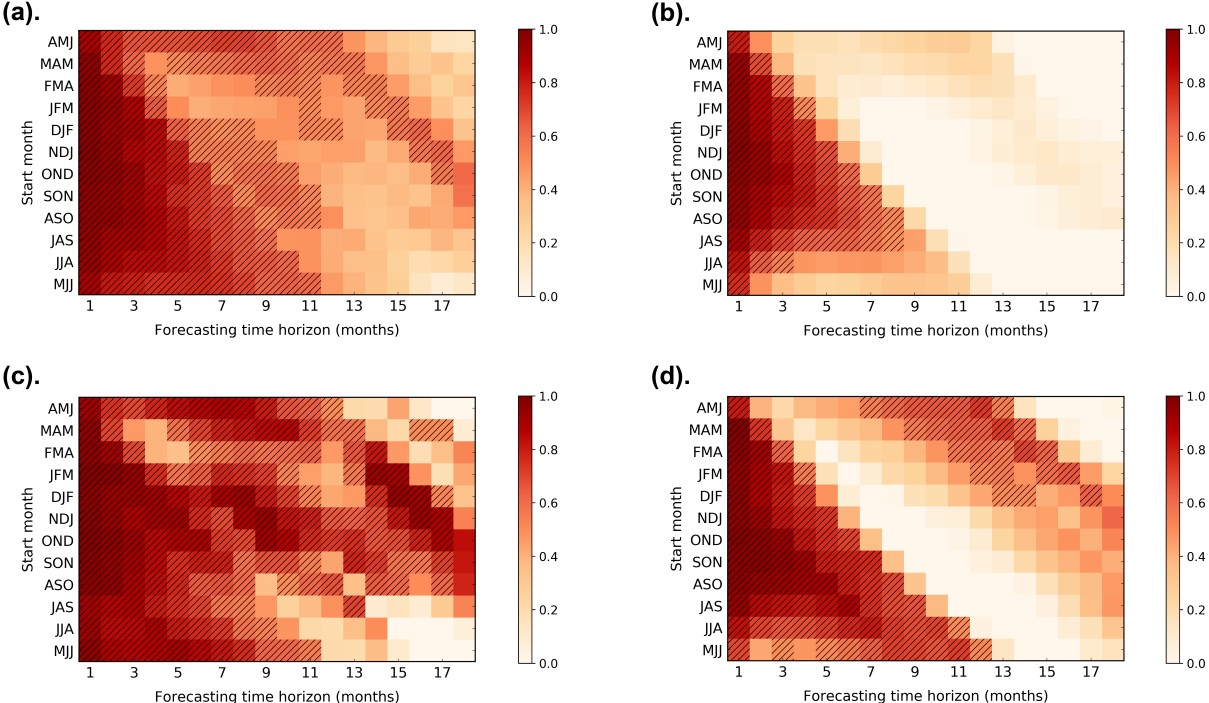

**Figure 5.** The correlation skills of the Nino3.4 index forecasts started from each calendar month in ENSO-MC using multi-step forecast strategy (a) and one-step ahead forecast strategy (b) for the GODAS data from 1982 to 2019. (c) is the same as (a), except for the GODAS data from 2010 to 2019. (b) is the same as (d), except for the GODAS data from 2010 to 2019. Hatches represent the forecasts with correlation skill exceeding 0.5.

for a lead of 11 months. And Figure 5(c) shows the results for the last decade whose validating period is from 2010 to 2019. The results in Fig. 5(c) perform only slightly skillful than those in Fig. 5(a), indicating the robustness of the model in terms of validation time. We also compare the two forecast strategies in deep learning, that is, the one-step ahead forecast strategy and the multi-step forecast strategy. Figures 5(a) and 5(c) are the results of multi-step prediction, while Figures 5(b) (1982-2019) and 5(d) (2010-2019) are the results of one-step prediction. It indicates regardless of the season from which the forecast is started, the skills would be reduced for predictions targeting the late boreal spring (May–June–July, MJJ). However, as shown in Figs. 5(b) and 5(d), the forecasting skills drop rapidly after the boreal spring, which seriously affects the subsequent forecasts, while the method of the multi-step strategy can reduce the influence (Figs. 5(a), (c)). We can conclude that the ENSO-MC model using multi-step forecast strategy is less affected by spring predictability barriers.

It should be noted that the correlation coefficient skills of Nino3.4 index obtained by simulating the SST anomalies spatial distribution are not as high as that obtained by using deep neural network to predict the Nino3.4 index directly, such as the results of Ham et al. (2019). This is because the former has a higher prediction target dimension, and the cumulative error of



each point of the spatial field is larger than the error of direct prediction of a single index. This is also one of the key issues to
be solved in the future development of the ENSO-MC model.

## 4   Precursor analysis of two types of ENSO

Based on the ENSO-MC model that successfully simulates different types of ENSO events, we can further explore the ENSO
dynamics learned by the ENSO-MC model and observe the signals before the onset of events. Considering the important role
of subsurface thermal memories in ENSO prediction, the precursory characteristics in heat content of different types of events
are discussed here. We select five EP El Niño events (1986, 1992, 1997, 2006 and 2015), three CP El Niño events (1994, 2002
and 2009) and three La Niña events (1988, 2008 and 2010) occurred in the past 30 years, and calculate the precursor maps of
heat content anomaly in the year prior to each event.

Specifically, the precursor maps of each event are obtained by computing the gradient of the regressed output with respect
to the input with saliency map method. The maps tell us how the output value will change when the pixel at this position in the
input image changes slightly, that is, the sensitivity of the predicted results to the perturbations in each region. In this way, we
can get the initial perturbation distribution that would develop into an ENSO event. The gradient calculation is equivalent to
the process of seeking the gradient of the errors with respect to the weights during training a machine learning model, which
represents the contribution of each weight to the total loss. We calculate the deviations between the output $\hat{Y}$ and the real
situation $Y$, fix the weights of the ENSO-MC model and perform back propagation for each pixel $(x, y)$ in the input $H$ at each
moment $\tau$.

$$loss = \mathcal{L}(Y, \hat{Y}) \tag{6}$$

$$M_\tau = \frac{\partial loss}{\partial H_{x,y}^\tau} \tag{7}$$

Then we obtain a series of heat maps $M : M_1, \ldots, M_t$ over the heat content predictors $H_1, \ldots, H_t$. Each $M_\tau$ indicates the
perturbation sensitivity distribution in heat content $H_\tau$ for $\tau$-lead time. For each event, twelve heat maps $M_1, \ldots, M_{12}$ are
obtained, which describe the precursor development in the year preceding the event. We add up the maps of $\tau$-lead time of
each event for one type to obtain the composite evolution maps $M_1^{comp}, \ldots, M_{12}^{comp}$ for each type. For example, the composite
precursor map of $\tau$-month lead for EP-type El Niño is obtained by adding up the precursor maps of $\tau$-lead time for all five EP
events. The composite maps of CP-type El Niño and La Niña are obtained in the same way. The precursor maps from 12-month
lead to 1-month lead of EP-type El Niño, CP-type El Niño and La Niña are shown in the Fig. 6, and we present results every
few months for each type to see more clearly how precursors change over time.

Figures 6(a), (b) show the heat content precursors of the EP-type El Niño events and CP-type El Niño events, respectively.
On the whole, the subsurface signals distributed in Fig. 6(a) are more intense and more extensive than those in Fig. 6(b),





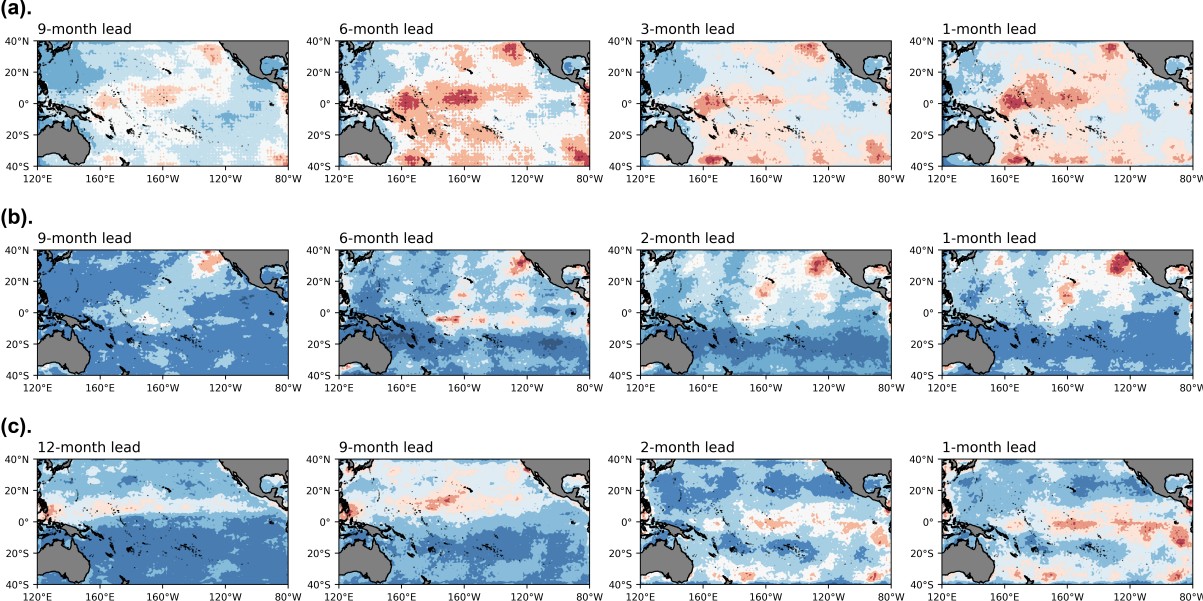

**Figure 6.** The composite evolution maps of initial perturbations in heat content before (a) EP-type El Niños, (b) CP-type El Niños and (c) La Niñas from 12-month lead to 1-month lead.

indicating that the occurrence of the EP-type El Niño is more related to the subsurface dynamics, while the CP events may be more affected by the atmospheric convection. Specifically, compared with Fig. 6(b) (CP-type El Niño), Fig. 6(a) (EP-type El Niño) shows a more pronounced signal, especially in the equatorial Pacific. It may be related to the stronger zonal tilt change of the equatorial thermocline and larger eastward movement of convection in tropical Pacific before the EP-type events. While the equatorial subsurface signal is weak in Fig. 6(b), but there is an obvious signal in the North Pacific. The results are consistent

with the previous studies that the negative phase of the North Pacific Oscillation promotes the development of SST anomalies in the central Pacific (Yu and Kim, 2011). Besides, there are robust signals over the northeastern Pacific in both types of El Niño (Figs. 6(a), (b)). The distribution is similar to the spatial structure of the Pacific meridional mode (PMM). PMM is forced by mid-latitude atmospheric variability in the Northern Hemisphere and evolves equatorward subsequently, which can affect ENSO. It indicates that signals outside the tropics play an important role in the prediction of El Niño and PMM can be regarded

as a precursor to El Niño. For La Niña events shown in Fig. 6(c), there are precursor signals propagating eastward from the western tropical Pacific in the subsurface from 12-month lead to the occurrence. Combined with the mechanism of the La Niña event, the signal would shoal the thermocline in the eastern Pacific and enhance the upwelling of cold subsurface waters, thereby ending the El Niño event and triggering a subsequent cold event.



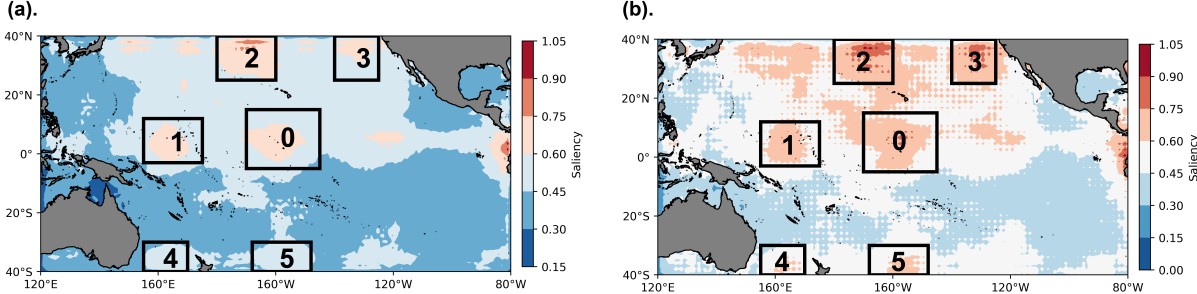

**Figure 7.** The composite saliency maps of surface (a) and subsurface (b) over $40°N - 40°S$, $120°E - 80°W$. The higher the saliency value, the more sensitive the area. Since the saliency map results obtained from the surface layer and the subsurface layer may affect each other, we first select six common areas with large values as candidates to perform sensitivity experiments. The six areas are the central equatorial Pacific (0), the western equatorial Pacific (1), the northern Central Pacific (2), the northeastern Pacific (3), the southwestern Pacific (4) and the southern Central Pacific (5).

## 5 Targeted observation sensitive area identification of ENSO

A saliency map shows which input pixels produce the largest increase in output values with minimal change. The idea is similar to the targeted observation strategy, that is, prioritizing the deployment of observations in the sensitive areas where small perturbations tend to have the greatest impacts on the forecasts. The saliency map method is therefore appropriate to identify the sensitive areas of targeted observation for ENSO. The areas with large values in the saliency map indicate that improving the accuracy of observations in these sensitive areas is the most efficient way to correct the output. Here we explore

the sensitive areas of the surface and the subsurface layer respectively, so in addition to heat content, the saliency maps of SST are also calculated. Since the sensitive area is a common attribute, it should be universal to all ENSO events and different types of ENSO are not considered here. We select eight El Niño events (1986, 1992, 1994, 1997, 2002, 2006, 2009 and 2015) and three La Niña events (1988, 2008 and 2010) that occurred in the past 30 years, and calculate the saliency maps of SST and heat content for each event according to the method described in Sect. 4. Then all the saliency maps of SST are added

up to obtain the composite saliency map of the surface (Fig. 7(a)), and that of the subsurface (Fig. 7(b)) is obtained in the same way. Since ENSO-MC regards multi-variable fields as multiple channels, the saliency regions obtained from surface layer and subsurface layer may affect each other. As shown in Fig. 7, the large-value region of surface (a) overlaps with that of subsurface (b). Therefore, we first artificially define six common regions with large values (black boxes in the Fig. 7), and perform sensitive experiments on the surface and subsurface respectively. Since the error is random in real observations, we use

random perturbations for sensitivity experiments. For each of the 11 selected ENSO events, 30 sets of random perturbations are superimposed to the original input field. The experiment that superimposes whole-field perturbations is called "all_rand", and the experiment that removes perturbations in the target area based on the whole-field perturbations is called "remove_rand". For the six regions shown in Fig. 7, the sensitivity of each region is measured according to Eq. (8), that is, the reduction of





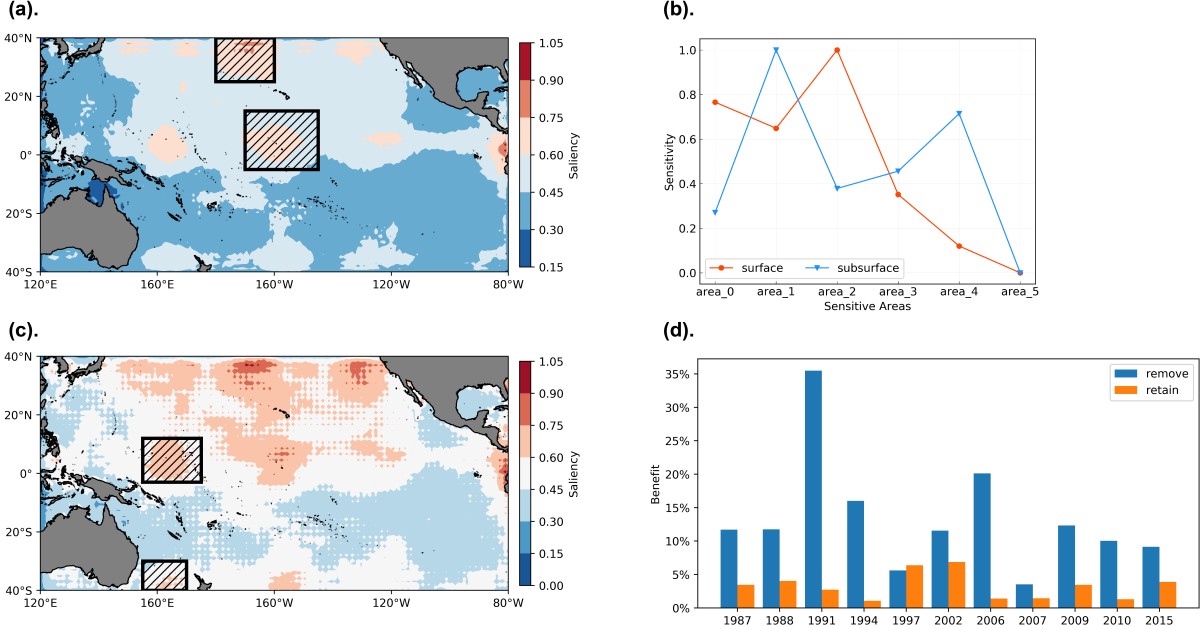

**Figure 8.** Sensitive areas identification results for ENSO with the saliency map method. (b) compares the sensitivity of the six candidate areas (0-5) for surface (red) and subsurface (blue). The first two most sensitive areas are selected as the sensitive areas of targeted observation for ENSO, that is, the area_0 and area_2 for surface and the area_1 and area_4 for subsurface. Specifically, the hatching areas in (a) (surface) and (c) (subsurface). In order to evaluate the effectiveness of the above identified sensitive areas, (d) compares the benefit for removing the random perturbations in the sensitive areas (blue) and removing the random perturbations outside the sensitive areas (orange, that is, retaining the random perturbations in the sensitive areas) for the eight El Niño events and three La Niña events occurred in the past 30 years.

prediction error caused by removing random perturbations in each region.

$$Sensitivity = \frac{\sum_i [\mathcal{L}_{RMSE}(Y_i, \hat{Y}_i^{all}) - \mathcal{L}_{RMSE}(Y_i, \hat{Y}_i^{remove})]}{\sum_i \mathcal{L}_{RMSE}(Y_i, \hat{Y}_i^{all})} \tag{8}$$

For each event $i$, $\hat{Y}_i^{all}$ represents the prediction results of experiment "all_rand", $\hat{Y}_i^{remove}$ represents the prediction results of experiment "remove_rand", and $Y_i$ represents the real observation. The results are shown in Fig. 8(b). The surface areas with the highest sensitivity for ENSO are No.0 and No.2 areas, that is, the central equatorial Pacific and the northern Central Pacific shown in Figure 8a, while the subsurface areas with the highest sensitivity are No.1 and No.4 areas, namely the western equatorial Pacific and the southwestern Pacific shown in the Fig. 8(c).

Furthermore, we perform two sets of experiments to measure the benefit of effective observations in the identified sensitive areas for improving forecast results. The first set calculates the reduction of the prediction error after removing the random perturbations in the identified sensitive areas ("remove"), and the second calculates the reduction after removing the perturbations outside the sensitive areas, that is, there are perturbations in the hatching areas in Figs. 8(a), (c) ("retain"). For each ENSO event, we superimpose 30 groups of random perturbations on the original input field, select the random perturbations





$k$ whose errors are larger than the mean error of all groups, and then conduct experiment "remove" and experiment "retain" respectively. Then we calculate the benefit $B_{remove}$ and $B_{retain}$, where $V_{in}$ is the volume of the identified sensitive areas, $V_{out}$ is the volume outside the sensitive areas.

$$B_{remove} = \frac{\sum_k [\mathcal{L}_{RMSE}(Y_k, \hat{Y}_k^{all}) - \mathcal{L}_{RMSE}(Y_k, \hat{Y}_k^{remove})]}{V_{in} \sum_k \mathcal{L}_{RMSE}(Y_k, \hat{Y}_k^{all})} \tag{9}$$


$$B_{retain} = \frac{\sum_k [\mathcal{L}_{RMSE}(Y_k, \hat{Y}_k^{all}) - \mathcal{L}_{RMSE}(Y_k, \hat{Y}_k^{retain})]}{V_{out} \sum_k \mathcal{L}_{RMSE}(Y_k, \hat{Y}_k^{all})} \tag{10}$$

$B_{remove}$ represents the degree of reduction of prediction errors after implementing target observation in per volume of identified sensitive areas shown in Figs. 8(a), (c), and $B_{retain}$ represents the reduction after implementing target observation in per volume of non-sensitive areas (here is the outside areas of hatching regions in Figs. 8(a), (c)). After removing the perturbations inside or outside the sensitive areas, the prediction errors are both reduced. However, due to the small size of sensitive areas, the benefit obtained by removing random perturbations in the sensitive areas is relatively high (Fig. 8(d)), except for the super strong El Niño event in 1997. Therefore, it is reasonable to give priority to effective observations in these identified sensitive areas, which are located in the central equatorial Pacific and the northern Central Pacific surface region, and western equatorial Pacific and the southwestern Pacific subsurface region. The results for the equatorial region support the conclusions of Kumar et al. (2014) and Duan and Hu (2016) in previous studies. Kumar et al. suggested that the observations in the central Pacific are more crucial than those in the eastern Pacific because of their role in preserving the memory of ENSO evolution. Duan and Hu emphasized the importance of subsurface signals in the western Pacific for ENSO predictions, which can influence the surface through equatorial waves and thermodynamic effects. However, due to the complexity of different models has a great impact on the identification of sensitive areas, there is no consistent conclusion about the sensitive areas of ENSO at present. For example, based on the outputs of CMIP5 model, Zhang et al. (2015) identified the central-eastern equatorial surface region and eastern subsurface region as the sensitive areas. Therefore, determining the most appropriate regions for target observation remains to be a long-standing challenge. Nevertheless, such studies based on interpretability can improve our understanding of how the ENSO-MC model works in ENSO prediction. The results of sensitive area identification support the theoretical understanding that oceanic thermal anomaly in the central and western Pacific provides a key long-term memory for SST predictions. In addition, the results show that processes outside the tropical Pacific also have an impact on ENSO prediction, such as surface temperature variations in the northern Central Pacific and subsurface thermal changes in the southwestern Pacific.

## 6 Conclusions

With the successful application of deep learning algorithms in ENSO forecasts, this paper attempts to expand the application scope of deep neural networks from prediction to a broader field, including the pattern simulation, understanding and observation of ENSO. For reliable forecasts of the two types of ENSO, a multichannel data-driven model ENSO-MC is proposed to




simulate the diversity of spatial patterns during ENSO events. Based on the ENSO-MC model, we then provide a new promising approach to investigate the early signals of different types of ENSO events and identify the sensitive areas with the help of saliency map interpretability method.

Specifically, the model ENSO-MC driven with oceanic and atmospheric predictors is proposed to simulate the ocean-atmosphere coupling process and predict the changes in spatial distribution of sea surface temperature anomalies. The simulation results show that the model can predict the development of SST anomalies in the equatorial Pacific during the onset, growth, maturity and decay of the El Niño and La Niña events one year in advance. In particular, we simulate the changes in the SST anomalies field of typical 2015/16 EP-type El Niño and 2009 CP-type El Niño at lead time beyond one year. With the

SST pattern forecasts, the all-season correlation skill of the Nino3.4 index in the ENSO-MC model is also evaluated, which is above 0.5 for a lead of 11 months. The precursor maps reveal the different pronounced characteristics of the subsurface signals before EP-type El Niño, CP-type El Niño and La Niña events. The results indicate that the EP-type El Niño is more related to the tropical thermocline dynamics, and the subtropical precursors seem to favor the generation of the CP-type El Niño. Both types of events have pronounced precursory signals in the northeastern Pacific whose distribution is similar to PMM. Before

the La Niña events, there is an obvious subsurface signal propagating eastward from the equatorial western Pacific, which would shoal the thermocline in the eastern Pacific and trigger a cold event. In addition, we present an attempt of the saliency method based on the ENSO-MC model for sensitive area identification of ENSO. The identification results show that the surface sensitive areas are located in the central equatorial Pacific and the northern Central Pacific, and the subsurface sensitive areas are concentrated in the western equatorial Pacific and the southwestern Pacific. Additional observations in these areas are

expected to better predict an event in the future. It indicates that in equatorial regions, the central surface area and the western subsurface area play an important role in the occurrence of future ENSO events, which are essential for preserving the memory of ENSO evolution. Besides, the processes in the extratropical Pacific also contribute to ENSO prediction, such as changes in the surface layer of the northern Central Pacific and the subsurface layer of the southwestern Pacific.

    Since the cumulative error of the Nino3.4 index calculated by predicting SST anomalies patterns is larger, the correlation

skill is not as high as that obtained by predicting the index directly. Further research should be undertaken to explore how to ensure high correlation skills of Nino3.4 index while correctly simulating the spatial distribution of SST anomalous. In addition to the existing components in the ENSO-MC model, how effectively use our existing domain knowledge, such as conservation of mass, conservation of salinity and other physical laws, to build physics-informed ENSO-MC model that may help reduce uncertainty and increase the credibility of predictions. For the precursor investigation, this paper focuses on verifying the

known ENSO mechanisms, and the unknown inherent characteristics exploration will be considered in the future. Combining structural causal model would help to extract the unknown causality relationship among factors and phenomena in ENSO complex interaction on this matter. This would help further explore the precursors of ENSO and improve our understanding of its predictability.



*Code and data availability.* The source code of the ENSO-MC 1.0 is available at https://doi.org/10.5281/zenodo.5725987. Thanks to Computational and Information Systems Laboratory (CISL) of NCAR, IRI/LDEO Climate Data Library, NOAA/OAR/ESRL PSL and ECMWF for providing the historical reanalysis data (https://rda.ucar.edu/) (Rayner et al., 2003); (https://iridl.ldeo.columbia.edu/) (Giese and Ray, 2011); (https://psl.noaa.gov/data/gridded/) (Reynolds et al., 2002; Behringer and Xue, 2004); (https://apps.ecmwf.int/datasets/) (Dee et al., 2011)

*Author contributions.* All authors designed the experiments and carried them out. Yuehan Cui developed the model code and performed the experiments. Yuehan Cui and Shijin Yuan prepare the manuscript with contributions from all co-authors.

*Competing interests.* The authors declare that they have no conflict of interest.

*Acknowledgements.* This study is supported in part by the key project fund of Shanghai 2020 "science and technology innovation action plan" for social development under Grant 20dz1200702, in part by the National Key Research and Development Program of China under Grant 2020YFA0608002, in part by the Fundamental Research Funds for the Central Universities under Grant 13502150039/003, and in part by the National Natural Science Foundation of China under Grant 42075141.





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
