# Peer review of "Simulation, Precursor Analysis and Targeted Observation Sensitive Area Identification for Two Types of ENSO using ENSO-MC v1.0"

_Geoscientific Model Development, 2021_

## Author Comment (AC3)

**Dear Editor and Reviewer:**

Thank you very much for your insightful comments concerning our manuscript "Simulation, Precursor Analysis and Targeted Observation Sensitive Area Identification for Two Types of ENSO using ENSO-MC v1.0" (ID: gmd-2021-396). Those comments are all very valuable and helpful for revising and improving our manuscript, and we have studied comments carefully and have made revisions. The manuscript has also been double-checked and modified, including the typos and grammar errors. The point-by-point responses are as following:

**Major Comments**

**Comment 1**

**Comment 1.1:** There are three parts of this manuscript, including ENSO prediction model named ENSO-MC, precursor analysis, and targeted observation sensitive area identification based on ENSO-MC. Accurately, the latter two parts are the application of ENSO-MC, and the conclusions are almost consistent with previous studies, such as Kumar et al. 2014, Duan and Hu 2016. Certainly, the method of precursor analysis and targeted observation sensitive area identification based on the deep neural network is an innovation, but I think the key point of this manuscript should be focusing on the ENSO-MC as the model description article. However, the description of ENSO-MC is inadequate, including the value of the weight in Loss function, discussions about the effects of the depth, structure, data for the neural network. I recommend authors add the discussion section.

**Response:** We gratefully thank you for the precious time the reviewer spent making constructive comments and suggestions on our manuscript. We also realize that the description of the model in the original manuscript is too brief. It only included a brief introduction of the neural network, data and loss function we used, and ignored the description of the idea of building the model for ENSO forecasting, the specific configuration and the effect of each component on the model performance.

As suggested by the reviewer, we have supplemented detailed descriptions of the ENSO-MC model structure, including its composition and hyperparameters. Specifically, the ENSO-MC model consists of an encoder, which is used to extract related features of ENSO, and a decoder, which is used to infer the development of SST anomalies in the equatorial Pacific in the future. And considering the important role of spatiotemporal interaction between atmosphere and ocean on ENSO, we use ConvLSTM as the main neural network architecture of encoder and decoder. In the encoder, each ConvLSTM layer followed by a 3D max-pooling layer is used to extract features at different spatial scales. Symmetric to the encoder structure, the decoder has upsampling layers followed by each ConvLSTM to restore the SST field. In addition, skip-layer connection and states connection, which make full use of the features extracted at each spatial scale of the encoder, are used to help the decoder recover the details of the forecast field. And the attention mechanism is designed to make the model automatically learn the temporal weights in the skip-layer connection. We also supplemented a figure (Fig 3(i)) in Section 3.3 to illustrate the skip-layer connection and its attention mechanism structure. For the hyperparameters in the model structure, the kernel size of ConvLSTM is  $3 \times 3$ , and the size of upsampling and downsampling is 2, which determines the field of feature extraction on the spatial scale. Besides, the depth of the model is 3, and the number of output filters of each layer is 8, 16, 32, which determines the degree of nonlinearity of the model.

Meanwhile, according to the variables, structure and loss function described in Section 2 of the manuscript, we also added a section to comprenhensively discuss the impact of these three factors on model performace, including the effects of the input sequence length, different predictors, each component of the model and different loss functions. Firstly, the input length represents the time dimension of ocean and atmosphere features that can be extracted by the model and plays an important role in ENSO forecast. In order to select the most suitable input length for ENSO-MC, we used the data of the past 3, 6, 9, 12 and 15 months as the input respectively, and compared the correlation skills and root mean square errors of their prediction results. Secondly, we designed an ablation experiment to verify the effectiveness of the multi-channel structure and examine the effect of each physical variable on the development of ENSO. Thirdly, we also supplemented experiments to prove the effectiveness of the skip-layer connection, states connection and attention mechanism structures for ENSO prediction, which are important components of model structure. Finally, for the combined loss functions, we detected the effect of each loss function by ablation experiment and determined the weight value of each function.

The detailed introduction about model structure and configurations have been supplemented in the Section 2.1 from the start of **line 105** as the **blue** text below:

Figure 1 The encoder-decoder architecture of ENSO-MC for SST pattern sequence prediction. The encoder contains three ConvLSTM layers and each layer is followed by a pooling layer, and the last layer is a convolutional layer, which allows extracting spatial and temporal features. And the decoder comprises one deconvolutional layer, three ConvLSTM layers and three upsampling layer that restores the features to the same size as the initial spatial dimension ( $80 \times 160$ ). The model uses skip-connection with attention mechanism and state-connection between the encoder and the decoder to improve forecasting skills. The input variables are the SST, heat content, zonal wind and meridional wind for  $T_{in}$  consecutive months over  $40^{\circ}N-40^{\circ}S$ ,  $120^{\circ}E-80^{\circ}W$  ( $80 \times 160$ ,  $1^{\circ} \times 1^{\circ}$  resolution) and each type of variable is input to the model as a channel. The output of the model is SST pattern sequence for the next  $T_{out}$  months.

"Here we develop a spatiotemporal model of multichannel structure named ENSO-MC to generate SST pattern sequence for ENSO forecasts. As shown in Fig. 1, the ENSO-MC is constructed based on the encoder-decoder architecture (Sutskever et al., 2014), whose encoder extracts the feature representations associated with ENSO over the past period and decoder generates the sea surface temperature pattern in the future. Due to the diversity of ENSO in amplitude, spatial pattern and temporal evolution, several convolutional long short-term memory (ConvLSTM) layers (Xingjian et al., 2015) form the skeleton in the encoder-decoder architecture to learn its multiple spatial and temporal representations. The encoder is the first half of the architecture (Fig. 1). A ConvLSTM layer with kernel size 3×3 followed by a 3D max-pooling layer constitutes an encoding module. The max-pooling layer downsamples the input along the spatial dimensions to extract multi-scale spatial connections. We use three encoding modules to construct the encoder, which is the network depth that perform best in ENSO forecasting problem here. To balance model performance and computational cost, the output channels of ConvLSTM in the three modules are 8, 16, 32 in order. After three encoding modules, we use a convolution layer with kernel size  $5 \times 5$  and stride  $5 \times 5$ , and the number of output channels is 64. The dimension of feature map output by each layer is shown in Fig. 1, and the final feature dimension of the encoder is  $2 \times 4 \times 64$ . The structure of the decoder is symmetrical with the encoder. After a transposed convolution layer, there are three decoding modules. Each module consists of an upsampling layer with size 2 followed by a ConvLSTM layer to restore the original resolution of SST pattern, where the kernel size of the network and the number of output channels are the same as those in the encoder. And the final layer in the ENSO-MC model is an additional 3×3 ConvLSTM that generates a single feature map representing the SST pattern sequence predicted by the model."

And the model evaluation for the input sequence length, different predictors, each model component and different loss functions have also been discussed in the Section 3. This new section has been added from the start of **line 205** as the blue text below:

**3 Model Performance evaluation**

**3.1 Influence of the input sequence length**

 Table 1: Correlation skill (Corr) and Root Mean Square Error (RMSE) of lead times with different input sequence lengths

| Sequence length | 3           | 6           | 9           | 12          | 15          |
|-----------------|-------------|-------------|-------------|-------------|-------------|
|                 | Corr/RMSE   | Corr/RMSE   | Corr/RMSE   | Corr/RMSE   | Corr/RMSE   |
| 3-month         | 0.78 / 0.67 | 0.59 / 1.02 | 0.58 / 1.24 | 0.81 / 0.54 | 0.68 / 1.10 |
| 6-month         | 0.52 / 0.96 | 0.46 / 1.15 | 0.47 / 1.16 | 0.64 / 0.81 | 0.48 / 1.72 |
| 12-month        | 0.29 / 1.00 | 0.36 / 1.39 | 0.28 / 1.14 | 0.53 / 0.84 | 0.31 / 2.06 |
| 18-month        | 0.17 / 1.06 | 0.25 / 1.72 | 0.15 / 1.17 | 0.44 / 0.91 | 0.30 / 2.15 |

Appropriate input sequence length is critical for ENSO prediction of the multichannel model. We use data from the past 3, 6, 9, 12 and 15 months as inputs to predict the development of ENSO in the next 18 months to examine the effects of different input sequence lengths on ENSO predictions. Table 1 shows the comparison of correlation skill and RMSE of lead times. For the correlation coefficient, the higher the value is, the higher the forecasting skill is. And the smaller RMSE represents the higher skill. The results show that the ENSO-MC model performs best with data from the past 12 months as input. This may be because the variables we select have long-term memory for the development of ENSO, such as oceanic heat content. A longer input sequence contains more information that is helpful to ENSO prediction, but also contains more noise that interferes with the prediction, so the improvement of forecasting skill is not positively correlated with the increase of input sequence length. But for different models and forecasting horizons, the most appropriate input sequence lengths may not be the same.

**3.2 Physical variable sensitivity for the multichannel structure**

Figure 2. The correlation skill of Nino 3.4 index of ENSO-MC model with different predictors.

With the physical variables we selected in the Section 2, we construct multichannel input that takes into account the complicated spatiotemporal interactions in the ocean and atmosphere underpinning the onset and development of ENSO events. For the grid observations of monthly SST, heat content, zonal wind and meridional wind, we treat each type of the input variable as a channel in the first ConvLSTM layer and thus there are four channels. The number of channels is the depth of the matrices involved in the convolutions, so that the cross-correlation and transmission between these ocean-atmosphere data can be calculated in the convolution operation. We design an ablation experiment to examine the contribution of predictors and the effectiveness of the multichannel structure. In addition to SST, the most important predictor, we remove heat content (t300) and wind from the inputs respectively to detect their effects on the correlation skill of the Nino3.4 index. As shown in Fig. 2, the model using the three key ingredients of Bjerknes feedback (SST, heat content, wind) as input produces more favorable forecast skill than the ones that remove one of them, which indicates that the multichannel structure can help to learn the ocean-atmosphere coupling between several important predictors. For the models with two predictors, the model containing wind predictor shows slightly higher forecasting skill in the first few months, while the one containing heat content predictor performs more stable skill at lead times of more than eight months. It suggests that the memory of subsurface heat plays an important role in ENSO prediction on seasonal to interannual time scales, which is consistent with previous research.

**3.3 Effectiveness of the model components**

Figure 3. (i) The detailed structure of the skip-layer connection and attention mechanism between encoder and decoder at the  $n^{th}$  layer in ENSO-MC. (ii) The correlation skill of Nino 3.4 index of the forecast lead month in models with different structures: (a) ENSO-MC model of skip-layer connection with attention mechanism and states connection, (b) ENSO-MC model without attention mechanism, (c) ENSO-MC model without states connection, (d) ENSO-MC model without skip-layer connection.

The ENSO-MC model learns the feature of ENSO at different spatial scales with the convolution and max-pooling layers in the encoder, and gradually restores the spatial dimensionality of the original SST field in the decoder. With symmetrical structure design of the encoder and decoder as shown in Fig. 1, skip-layer connection is used to transfer features form the encoder to the decoder to recover spatial information lost during downsampling (yellow line in Fig. 1). Rather than transferring the original features of all time steps obtained from the encoder, we design an attention mechanism to enable the skip-layer to automatically learn the attention weights  $\beta_1, \beta_2, ..., \beta_t$  on the temporal sequence because these air-sea features may have different effects on ENSO development at different time scales. As shown in Fig. 3(i), the encoder obtains the features  $f_n \in \mathbb{R}^{T_{in} \times h_n \times w_n \times c_n}$  after max-pooling and convolution calculation at the  $n^{th}$  layer. Using a two-layer densely-connected neural network, we obtain the attention weight  $\beta \in \mathbb{R}^{T_{in}}$  of each time step's features according to Eq. (1), where  $f'_n \in \mathbb{R}^{T_{in} \times (h_n \times w_n \times c_n)}$  are reshaped from  $f_n$ :

$$\beta = \operatorname{softmax}(\mathbf{W}_{\beta\alpha} \operatorname{tanh}(\mathbf{W}_{\alpha f} f'_{n} + \mathbf{b}_{\alpha f}) + \mathbf{b}_{\beta\alpha}), \tag{1}$$

where  $\mathbf{W}_{\alpha f}$ ,  $\mathbf{W}_{\beta \alpha}$  are weight matrices created by the layer, and  $\mathbf{b}_{\alpha f}$ ,  $\mathbf{b}_{\beta \alpha}$  are the bias vectors.  $\beta$  represents the contribution of each time step to prediction. According to Eq. (2), the feature maps of each time step are multiplied by the corresponding weights, and the fused maps  $\tilde{f}_n \in \mathbb{R}^{h_n \times w_n \times c_n}$  are obtained by adding them along the time dimension.

$$\bar{f}_n = \sum_{T_{in}} (\beta \circ f_n), \tag{2}$$

where  $\tilde{f}_n$  are the feature maps to be transmitted in the skip-layer connection, which are connected to the features of the corresponding layer in the decoder. Besides, we also add states connection between the encoder and the decoder (grey line in Fig. 1), where the hidden states output by the ConvLSTM layers in the encoder are reserved for the corresponding layer when the decoder is initialized. With the methods of skip-layer connection and states connection, the model can make full use of the information extracted from the encoder before ENSO events, which help stabilize training and convergence.

We remove the attention mechanism (model b), states connection structure (model c) and skiplayer connection structure (model d) respectively from the constructed ENSO-MC model (model a) to analyze their effects on model performance. As shown in Fig. 3(ii), the two connection structures, especially the skip-layer connection structure, have a great influence on the prediction results. In model b, we use average weights to replace the attention mechanism. Compared with model a, self-attention mechanism can play a greater role in long-term ENSO prediction.

---

## Author Comment (AC4)

Dear Editor and Reviewer:

Thank you very much for your insightful comments concerning our manuscript "Simulation, Precursor Analysis and Targeted Observation Sensitive Area Identification for Two Types of ENSO using ENSO-MC v1.0" (ID: gmd-2021-396). Those comments are all very valuable and helpful for revising and improving our manuscript, and we have studied comments carefully and have made revisions. The manuscript has also been double-checked and modified, including the typos and grammar errors. The point-by-point responses are as following:

**Major Comments**

**Comment 1:** Based on deep neural network, an ENSO prediction model with multiple physical variables is constructed in this manuscript to simulate the changes of SSTA, analyze the precursors and identify the sensitive areas in the equatorial Pacific. It is a good attempt to broaden the application field of neural network in climate research. And the description of the prediction model should be complete enough to reproduce. However, in the second section of the paper, the description of ENSO-MC model is somewhat brief. I suggest the authors add descriptions of specific configurations through experiments, such as how to determine the model hyperparameters.

**Response:** We gratefully appreciate your valuable suggestion. In the original manuscript, it only included a brief introduction of the neural network, data and loss function we used, which is indeed inadequate for the ENSO-MC model description.

As suggested by the reviewer, we have supplemented detailed descriptions of the ENSO-MC model structure, including its composition and hyperparameters. Specifically, the ENSO-MC model consists of an encoder, which is used to extract related features of ENSO, and a decoder, which is used to infer the development of SST anomalies in the equatorial Pacific in the future. And considering the important role of spatiotemporal interaction between atmosphere and ocean on ENSO, we use ConvLSTM as the main neural network architecture of encoder and decoder. In the encoder, each ConvLSTM layer followed by a 3D max-pooling layer is used to extract features at different spatial scales. Symmetric to the encoder structure, the decoder has upsampling layers followed by each ConvLSTM to restore the SST field. In addition, skip-layer connection and states connection, which make full use of the features extracted at each spatial scale of the encoder, are used to help the decoder recover the details of the forecast field. And the attention mechanism is designed to make the model automatically learn the temporal weights in the skip-layer connection. We also supplemented a figure (Fig 3(i)) in Section 3.3 to illustrate the skip-layer connection and its attention mechanism structure. For the hyperparameters in the model structure, the kernel size of ConvLSTM is 3×3, and the size of upsampling and downsampling is 2, which determines the field of feature extraction on the spatial scale. Besides, the depth of the model is 3, and the number of output filters of each layer is 8, 16, 32, which determines the degree of nonlinearity of the model.

Meanwhile, according to the variables, structure and loss function described in Section 2 of the manuscript, we also added a section to comprehensively discuss the impact of these three factors on model performace, including the effects of the input sequence length, different predictors, each component of the model and different loss functions. Firstly, the input length represents the time dimension of ocean and atmosphere features that can be extracted by the model and plays an

important role in ENSO forecast. In order to select the most suitable input length for ENSO-MC, we used the data of the past 3, 6, 9, 12 and 15 months as the input respectively, and compared the correlation skills and root mean square errors of their prediction results. Secondly, we designed an ablation experiment to verify the effectiveness of the multi-channel structure and examine the effect of each physical variable on the development of ENSO. Thirdly, we also supplemented experiments to prove the effectiveness of the skip-layer connection, states connection and attention mechanism structures for ENSO prediction, which are important components of model structure. Finally, for the combined loss functions, we detected the effect of each loss function by ablation experiment and determined the weight value of each function.

The detailed introduction about model structure and configurations has been supplemented in the Section 2 from the start of **line 105** as the blue text below:

[Figure]

*Figure 1 The encoder-decoder architecture of ENSO-MC for SST pattern sequence prediction. The encoder contains three ConvLSTM layers and each layer is followed by a pooling layer, and the last layer is a convolutional layer, which allows extracting spatial and temporal features. And the decoder comprises one deconvolutional layer, three ConvLSTM layers and three upsampling layer that restores the features to the same size as the initial spatial dimension (80×160). The model uses skip-connection with attention mechanism and state-connection between the encoder and the decoder to improve forecasting skills. The input variables are the SST, heat content, zonal wind and meridional wind for $T_{in}$ consecutive months over 40ºN-40ºS, 120ºE-80ºW (80×160, 1º×1º resolution) and each type of variable is input to the model as a channel. The output of the model is SST pattern sequence for the next $T_{out}$ months.*

"*Here we develop a spatiotemporal model of multichannel structure named ENSO-MC to generate SST pattern sequence for ENSO forecasts. As shown in Fig. 1, the ENSO-MC is constructed based on the encoder-decoder architecture (Sutskever et al., 2014),* whose encoder extracts the feature representations associated with ENSO over the past period and decoder generates the sea

surface temperature pattern in the future. Due to the diversity of ENSO in amplitude, spatial pattern and temporal evolution, several convolutional long short-term memory (ConvLSTM) layers (Xingjian et al., 2015) form the skeleton in the encoder-decoder architecture to learn its multiple spatial and temporal representations. The encoder is the first half of the architecture (Fig. 1). A ConvLSTM layer with kernel size 3×3 followed by a 3D max-pooling layer constitutes an encoding module. The max-pooling layer downsamples the input along the spatial dimensions to extract multi-scale spatial connections. We use three encoding modules to construct the encoder, which is the network depth that perform best in ENSO forecasting problem here. To balance model performance and computational cost, the output channels of ConvLSTM in the three modules are 8, 16, 32 in order. After three encoding modules, we use a convolution layer with kernel size 5×5 and stride 5×5, and the number of output channels is 64. The dimension of feature map output by each layer is shown in Fig. 1, and the final feature dimension of the encoder is 2×4×64. The structure of the decoder is symmetrical with the encoder. After a transposed convolution layer, there are three decoding modules. Each module consists of an upsampling layer with size 2 followed by a ConvLSTM layer to restore the original resolution of SST pattern, where the kernel size of the network and the number of output channels are the same as those in the encoder. And the final layer in the ENSO-MC model is an additional 3×3 ConvLSTM that generates a single feature map representing the SST pattern sequence predicted by the model."

And the model evaluation for the input sequence length, different predictors, each model component and different loss functions have also been discussed in the Section 3. This new section has been added from the start of **line 205** as the blue text below:

**3 Model Performance evaluation**

**3.1 Influence of the input sequence length**

**Table 1: Correlation skill (Corr) and Root Mean Square Error (RMSE) of lead times with different input sequence lengths**

| Sequence length | 3 | 6 | 9 | 12 | 15 |
|---|---|---|---|---|---|
| | Corr/RMSE | Corr/RMSE | Corr/RMSE | Corr/RMSE | Corr/RMSE |
| **3-month** | 0.78 / 0.67 | 0.59 / 1.02 | 0.58 / 1.24 | **0.81 / 0.54** | 0.68 / 1.10 |
| **6-month** | 0.52 / 0.96 | 0.46 / 1.15 | 0.47 / 1.16 | **0.64 / 0.81** | 0.48 / 1.72 |
| **12-month** | 0.29 / 1.00 | 0.36 / 1.39 | 0.28 / 1.14 | **0.53 / 0.84** | 0.31 / 2.06 |
| **18-month** | 0.17 / 1.06 | 0.25 / 1.72 | 0.15 / 1.17 | **0.44 / 0.91** | 0.30 / 2.15 |

Appropriate input sequence length is critical for ENSO prediction of the multichannel model. We use data from the past 3, 6, 9, 12 and 15 months as inputs to predict the development of ENSO in the next 18 months to examine the effects of different input sequence lengths on ENSO predictions. Table 1 shows the comparison of correlation skill and RMSE of lead times. For the correlation coefficient, the higher the value is, the higher the forecasting skill is. And the smaller RMSE represents the higher skill. The results show that the ENSO-MC model performs best with data from the past 12 months as input. This may be because the variables we select have long-term memory for the development of ENSO, such as oceanic heat content. A longer input sequence contains more information that is helpful to ENSO prediction, but also contains more noise that

interferes with the prediction, so the improvement of forecasting skill is not positively correlated with the increase of input sequence length. But for different models and forecasting horizons, the most appropriate input sequence lengths may not be the same.

**3.2 Physical variable sensitivity for the multichannel structure**

[Figure]

**Figure 2.** The correlation skill of Nino 3.4 index of ENSO-MC model with different predictors.

With the physical variables we selected in the Section 2, we construct multichannel input that takes into account the complicated spatiotemporal interactions in the ocean and atmosphere underpinning the onset and development of ENSO events. For the grid observations of monthly SST, heat content, zonal wind and meridional wind, we treat each type of the input variable as a channel in the first ConvLSTM layer and thus there are four channels. The number of channels is the depth of the matrices involved in the convolutions, so that the cross-correlation and transmission between these ocean-atmosphere data can be calculated in the convolution operation. We design an ablation experiment to examine the contribution of predictors and the effectiveness of the multichannel structure. In addition to SST, the most important predictor, we remove heat content (t300) and wind from the inputs respectively to detect their effects on the correlation skill of the Nino3.4 index. As shown in Fig. 2, the model using the three key ingredients of Bjerknes feedback (SST, heat content, wind) as input produces more favorable forecast skill than the ones that remove one of them, which indicates that the multichannel structure can help to learn the ocean-atmosphere coupling between several important predictors. For the models with two predictors, the model containing wind predictor shows slightly higher forecasting skill in the first few months, while the one containing heat content predictor performs more stable skill at lead times of more than eight months. It suggests that the memory of subsurface heat plays an important role in ENSO prediction on seasonal to interannual time scales, which is consistent with previous research.

**3.3 Effectiveness of the model components**

[Figure]

**Figure 3.** (i) The detailed structure of the skip-layer connection and attention mechanism between encoder and decoder at the $n^{th}$ layer in ENSO-MC. (ii) The correlation skill of Nino 3.4 index of the forecast lead month in

models with different structures: (a) ENSO-MC model of skip-layer connection with attention mechanism and states connection, (b) ENSO-MC model without attention mechanism, (c) ENSO-MC model without states connection, (d) ENSO-MC model without skip-layer connection.

The ENSO-MC model learns the feature of ENSO at different spatial scales with the convolution and max-pooling layers in the encoder, and gradually restores the spatial dimensionality of the original SST field in the decoder. With symmetrical structure design of the encoder and decoder as shown in Fig. 1, skip-layer connection is used to transfer features form the encoder to the decoder to recover spatial information lost during downsampling (yellow line in Fig. 1). Rather than transferring the original features of all time steps obtained from the encoder, we design an attention mechanism to enable the skip-layer to automatically learn the attention weights $\beta_1, \beta_2, \ldots, \beta_t$ on the temporal sequence because these air-sea features may have different effects on ENSO development at different time scales. As shown in Fig. 3(i), the encoder obtains the features $f_n \in \mathbb{R}^{T_{in} \times h_n \times w_n \times c_n}$ after max-pooling and convolution calculation at the $n^{th}$ layer. Using a two-layer densely-connected neural network, we obtain the attention weight $\beta \in \mathbb{R}^{T_{in}}$ of each time step's features according to Eq. (1), where $f_n' \in \mathbb{R}^{T_{in} \times (h_n \times w_n \times c_n)}$ are reshaped from $f_n$:

$$\beta = \text{softmax}\big(\mathbf{W}_{\beta\alpha} \tanh\big(\mathbf{W}_{\alpha f} f_n' + \mathbf{b}_{\alpha f}\big) + \mathbf{b}_{\beta\alpha}\big), \tag{1}$$

where $\mathbf{W}_{\alpha f}$, $\mathbf{W}_{\beta\alpha}$ are weight matrices created by the layer, and $\mathbf{b}_{\alpha f}$, $\mathbf{b}_{\beta\alpha}$ are the bias vectors. $\beta$ represents the contribution of each time step to prediction. According to Eq. 2, the feature maps of each time step are multiplied by the corresponding weights, and the fused maps $\widetilde{f_n} \in \mathbb{R}^{h_n \times w_n \times c_n}$ are obtained by adding them along the time dimension.

$$\widetilde{f_n} = \sum_{T_{in}} (\beta \circ f_n), \tag{2}$$

where $\widetilde{f_n}$ are the feature maps to be transmitted in the skip-layer connection, which are connected to the features of the corresponding layer in the decoder. Besides, we also add states connection between the encoder and the decoder (grey line in Fig. 1), where the hidden states output by the ConvLSTM layers in the encoder are reserved for the corresponding layer when the decoder is initialized. With the methods of skip-layer connection and states connection, the model can make full use of the information extracted from the encoder before ENSO events, which help stabilize training and convergence.

We remove the attention mechanism (model b), states connection structure (model c) and skip-layer connection structure (model d) respectively from the constructed ENSO-MC model (model a) to analyze their effects on model performance. As shown in Fig. 3(ii), the two connection structures, especially the skip-layer connection structure, have a great influence on the prediction results. In model b, we use average weights to replace the attention mechanism. Compared with model a, self-attention mechanism can play a greater role in long-term ENSO prediction.

**3.4 Effects of different loss functions**

[Figure]

**Figure 4.** The performances of the ENSO-MC with different loss functions.

In order to balance the optimization speed of each loss in the training process, we set $\lambda_{mse}$=7, $\lambda_{ssim}$ = 9 and $\lambda_{gdl}$=1. The effectiveness of combined loss function is validated. As shown in Fig. 4(a), although SSIM and GDL do not significantly improve the model performance when combined with MSE alone, the combination of MSE, SSIM and GDL loss functions achieve the best performance on the correlation skill. Besides, since GDL loss function tends to retain extreme values and MSE loss function tends to smooth all values, the presence of GDL inhibits the decrease of MSE, so the MSE errors of the models with GDL loss function are higher than the ones without GDL (Fig. 4(b)). And comparing the results of correlation skill and RMSE in Fig. 4(a) and (b), low RMSE values do not represent high correlation skills. Therefore, it is necessary to explore loss functions suitable for ENSO prediction other than MSE to balance the training of the model.

**Comment 2:** For the prediction results of EP, CP and La Nina events, the authors select one case to analyze respectively in the third section. It is suggested to add cases or discuss the overall forecasting performance of different types of ENSO, for example, the forecast results of different types of events in the validation set can be summarized into a table.

**Response:** Thank you for your rigorous consideration and professional comments. In the original manuscript, we only selected three individual events of 2015/2016 EP El Niño, 2009/2010 CP El Niño and 1988/1989 La Niña to validate the forecast skills in spatial patterns and Niño 3.4 index time series. The simulation ability of ENSO-MC model for different types of events is not fully explained in the manuscript. As suggested by the reviewer, we have added the simulation results of three more cases in recent years for each type of event, namely, EP El Niño events of 1991/1992, 1997/1998 and 2006/2007, CP El Niño events of 1994/1995, 2002/2003 and 2018/2019, and La Niña events of 1984/1985, 1998/1999 and 2000/2001. We compare the predicted spatial patterns and observations of SST anomalies from the onset to the mature phase for these events. The simulation results show that ENSO-MC model can generally simulate the development and characteristics of SSTA for different types of events. We also summarized the classification results of the model for the two types of El Niño events occurring from 1984 to 2019, and calculated the RMSE of Niño3, Niño3.4 and Niño4 index for different lead months. The results show that ENSO-MC model has a higher classification accuracy for EP events, and smaller prediction errors for CP events in amplitude.

The related results and statements have been supplemented in the Section 4 from the start of

**line 304** as the blue text below:

"In addition to the above three typical events in recent years, the prediction results of other events that occurred between 1984 and 2019 are also detected. For each event, we compare the spatial development of predicted and observed SST anomalies in the equatorial Pacific from the onset to the maturity stage.

**(a).**

[Figure]

**Figure 5.** SSTAs of three EP El Niño events in (a) 1991/1992, (b) 1997/1998 and (c) 2006/2007 from the onset to the maturity stage, with observations in the first row and predictions in the second row for each event. The mature phases here are the months when the El Niño events peak. And "0" and "1" next to the calendar month denote the year when the El Niño event occurred and the following year, respectively.

Fig. 5 shows the simulation results of the ENSO-MC model for three EP El Niño events of 1991/1992, 1997/1998 and 2006/2007, with observations in the first row and predictions in the second row for each group. The results show that the model can simulate the occurrence and development of SSTA for each event. However, for some events with less significant EP type characteristics (for example, that of 1991/1992), the SSTA center of predictions is closer to the central Pacific than observed. In addition, for some super strong events (for example, that of

1997/1998) and weak events (for example, that of 2006-2007), the amplitude of the predicted results at mature phase may be lower or higher than the observed.

[Figure]

**Figure 6.** As in (Fig. 5, but for the three CP El Niño events in (a) 1994/1995, (b) 2002/2003 and (c) 2018/2019.

The prediction results for three CP El Niño events in 1994/1995, 2002/2003 and 2018/2019 are displayed in Fig. 6. For the events of 1994/1995 and 2018/2019, the model can simulate the process that the SST anomalies in the northeast Pacific propagate to the southwest and finally contribute to the occurrence of CP events. The amplitude and center location of the predicted anomalies are also in agreement with the observations. However, the meridional distribution of predicted SSTA is not as broad as observed in the mature stage. The observed SSTA extends eastward to 80ºW, while the predicted value extends roughly between 100ºW and 120ºW.

[Figure]

**Figure 7.** As in Fig. 5, but for the three La Niña events in (a) 1984/1985, (b) 1998/1999 and (c) 2000/2001.

And Fig. 7 shows the predictions of three La Niña events in 1984/1985, 1998/1999 and 2000/2001. These three events occurred under different conditions. The events of 1984/1985 and 1998/1999 were preceded by strong El Niño events, and the 1998/1999 event occurring more rapidly. The 2000/2001 La Niña was another weaker event after the previous La Niña event ended. Compared with observations, the model can simulate the occurrence, development and phase transition or persistence of La Niña events.

[Figure]

**Figure 8.** Scatterplots in Nino3-Nino4 index plane of 12-month-lead predictions for all (a) EP El Niño events and

(b) CP El Niño events during peak phase from 1984 to 2019. (c) Root Mean Square Error (RMSE) of the Nino3.4, Nino3 and Nino4 indexes between the forecast results of the ENSO-MC model and observations during validation period.

**Table 2: Root Mean Square Error (RMSE) of Nino3/4 index for all EP/CP El Niño events during peak phase from 1984 to 2019.**

| Lead time | 3-month | 6-month | 9-month | 12-month |
|---|---|---|---|---|
| EP (Nino3) | 1.18 | 1.23 | 1.04 | 1.07 |
| CP(Nino4) | 0.45 | 0.99 | 0.91 | 0.93 |

In addition to comparing the detailed spatial distribution of SSTA, the related indices and metrics are calculated to further evaluate the simulation performance of the ENSO-MC model. The Niño 3 index (average SST anomalies over 5ºN-5ºS, 150ºW-90ºW) and the Niño 4 index (average SST anomalies over 5ºN-5ºS, 160ºE-150ºW) are commonly used to define two types of El Niño events. Events with Niño 4 index greater than Niño 3 are regarded as CP El Niños, and events with Niño 3 index greater than Niño 4 are classified as EP El Niños. Figure 8a, b shows the distribution for Niño 3 and Niño 4 indexes calculated from the model's one-year-lead predictions of the peak periods for all EP events (Fig. 8a) and CP events (Fig. 8b) from 1984 to 2019. The results show that the model can correctly classify five EP events (1987/1988, 1991/1992, 1997/1998, 2006/2007, 2015/2016) and three CP events (1994/1995, 2002/2003, 2018/2019) in the past 30 years, but misjudge the event of 2009/2010 as EP type and no El Niño event occurred in 2004 (Niño 3=0, Niño 4=0). The CP event of 2004/2005 is much weaker than other CP ones, making it more difficult for the model to capture its development. We also make statistics on the RMSE between the predictions and official records of Nino3/4 index for all EP/CP El Niño events at mature phase (Table 2) and for the whole validation period (Fig. 8c), and find that although the model has a higher classification accuracy for EP events, the index error of predictions for EP events is larger than that for CP. It may be because most of the strong El Niños are EP-type events, and the prediction skills of the model for such extreme events need to be improved. The SSTA distribution in Fig. 5 also shows that for some EP events, there is a difference in amplitude between predictions and observations for the maturity stage of the event, while that of the CP events is consistent with the observations (Fig. 6)."

**Comment 3:** In the fourth section, based on the proposed ENSO-MC model, saliency map method is used to analyze the subsurface precursors. Since the changes of ENSO originate from the strong interactions between oceanic and atmospheric changes, it is recommended to analyze the changes of SST and wind field while analyzing the changes of precursors in heat content. For example, the process can be further elaborated according to the influence of ocean wave and Walker circulation on ENSO.

**Response:** Thank you so much for your professional attitude and insightful suggestion. Compared with only mentioning the characteristics of the subsurface precursors in the original manuscript, it makes the analysis more complete that further analysis of the air-sea process affecting ENSO by combining SST and wind field changes. We chose the 1997-1998 EP El Niño as a case, and analyzed the influence of westerly wind events and thermocline dynamics on the occurrence and development

of the event with longitude-time diagram. In addition, we selected two El Niño events of 1994-1995 and 2015-2016 to analyze the impact of PMM on the event through Wind-Evaporation-SST (WES) feedback. The results show that EP events are mostly related to the subsurface dynamics of the equatorial Pacific, and PMM sometimes contributes to the development of El Nino events, both CP and EP events.

The detailed analysis has been supplemented in the Section 5 from the start of **line 413** as the blue text below:

[Figure]

*Figure 9 The composite evolution maps of initial perturbations in heat content before (a) EP-type El Niños, (b) CP-type El Niños and (c) La Niñas from 12-month lead to 1-month lead.*

"*On the whole, the subsurface signals distributed in Fig. 9(a) are more intense and more extensive than those in Fig. 9(b), indicating that the occurrence of the EP-type El Niño is more related to the subsurface dynamics, while the CP events may be more affected by the atmospheric convection. Specifically, compared with Fig. 9(b) (CP-type El Niño), Fig. 9(a) (EP-type El Niño) shows a more pronounced signal, especially in the equatorial Pacific. It may be related to the stronger zonal tilt change of the equatorial thermocline and larger eastward movement of convection in tropical Pacific before the EP-type events.*

[Figure]

Fig. 10 Longitude-time diagram of monthly surface zonal wind anomalies (left), SST anomalies (middle), and heat content (t300) anomalies (right) across the equatorial Pacific (2°N-2°S, 120°E-80°W) from September 1996 to April 1998. Data are based on NCEP Global Ocean Data Assimilation System and ERA-5.

For example, as shown in Fig. 10, a series of westerly wind events along the equatorial Pacific led to an abrupt relaxation and reversal of trade winds in the western and central equatorial Pacific in early 1997. The westerly wind anomalies generated downwelling Kelvin waves, which propagated eastward and deepened the thermocline in the eastern Pacific in late 1997. The depressed thermocline limited the upwelling of subsurface cold water, prompting the development of warm surface temperatures. Meanwhile, westward-propagating Rossby waves shallowed the thermocline in the western Pacific. These processes led to significant changes in the equatorial thermocline (Fig. 9(a)), a flattening of the thermocline and a decrease in the zonal SST gradient along the equator. The reduction of the SST gradient in turn further weakened the trade winds, leading to the rapid development of the 1997/1998 El Niño. *La Niña events usually occur in the second year after a warm event. As shown in Fig. 9(c), there are precursor signals produced by wind forcing propagating eastward from the western tropical Pacific in the subsurface from 12-month lead to the occurrence. Combined with the mechanism of the La Niña event, the signal would shoal the thermocline in the eastern Pacific and enhance the upwelling of cold subsurface waters, thereby ending the El Niño event and triggering a subsequent cold event.*

[Figure]

Fig. 11 The SST and 10-m wind-vector anomalies for the different seasons before 1994/1995 CP Niño and 2015/2016 EP Niño.

*While the equatorial subsurface signal is weak in Fig. 9(b), there is an obvious signal in the North Pacific. The results are consistent with the previous studies that the negative phase of the North Pacific Oscillation promotes the development of SST anomalies in the central Pacific (Yu and Kim, 2011). Besides, there are robust signals over the northeastern Pacific in both types of El Niño (Figs. 9(a), (b)). The distribution is similar to the spatial structure of the Pacific meridional mode (PMM). PMM is forced by mid-latitude atmospheric variability in the Northern Hemisphere and evolves equatorward subsequently, which can affect ENSO.* As shown in Fig. 11, one year before the 1994/1995 CP El Niño, there were warm subtropical SST anomalies extending southwest from Baja California. The SST anomalies weakened the trade winds and reduced the surface evaporation over the region via Wind-Evaporation-SST (WES) feedback. The reduction in evaporation allowed warm waters to expand further southwestward, enhancing the PMM and eventually reaching the equator, which weakened equatorial trade winds and triggered an El Niño event in late 1994. PMM not only appeared before CP El Niño, for example, the emergence of PMM in late 2014 contributed to the development of 2015/2016 El Niño (Fig. 11). *It indicates that signals outside the tropics play an important role in the prediction of El Niño and PMM can be regarded as a precursor to El Niño.*"

**Minor Comments**
**Comment 1:** Lines 28-29: For the predictability study of two types of El Nino, Tian and Duan (2015) demonstrated that the spring predictability barrier is weaker in CP-El Nino than in EP-El Nino when model error effects can be negligible. Tao et al. (2020) used the nonlinear forcing singular vector (NFSV)-tendency assimilation approach to improve ENSO model and showed the ability of recognizing the types of El Nino at least six months in advance in predictions (also see Tao and Duan, 2019).

- Ben Tian and Wansuo Duan, Comparison of the initial errors most likely to cause a spring predictability barrier for two types of El Nino events, Clim Dyn, 2015, DOI:10.1007/s00382-015-2870-0
- Tao Lingjiang, and Wansuo, Duan, Using a nonlinear forcing singular vector approach to reduce model error effects in ENSO forecasting. Weather and Forecasting. 2019. 1321-1342. DOI: 10.1175/WAF-D-19-0050.1
- Tao Linjiang, Duan Wansuo, and Stephane Vannitsem, Improving forecasts of El Niño diversity: a nonlinear forcing singular vector approach. Climate Dynamics. 2020. 55: 739-754. doi:

10.1007/s00382-020-05292-5

**Response:** We sincerely appreciate the valuable comments. We have read the literature carefully and supplemented the related statements and references from the start of **line 29** as the **blue** text below:

"*In recent decades, with the increased occurrence of CP El Niño relative to EP El Niño, the predictability of two ENSO types has attracted widespread attentions (Lee and McPhaden, 2010).* **Tao et al. (2020) used the nonlinear forcing singular vector (NFSV)-tendency assimilation approach to improve ENSO model and showed the ability of recognizing the types of El Niño at least six months in advance in predictions (Lingjiang and Wansuo, 2019). Tian and Duan (2016) demonstrated that the spring predictability barrier is weaker in CP-El Niño than in EP-El Niño when model error effects can be negligible.** *Improved forecasting and understanding of the two types of ENSO are therefore of great importance.*"

**References**

**Tao, L., Duan, W., and Vannitsem, S.: Improving forecasts of El Niño diversity: a nonlinear forcing singular vector approach, Climate Dynamics, 55, 739–754, 2020.**

**Lingjiang, T. andWansuo, D.: Using a nonlinear forcing singular vector approach to reduce model error effects in ENSO forecasting, Weather and Forecasting, 34, 1321–1342, 2019.**

**Tian, B. and Duan, W.: Comparison of the initial errors most likely to cause a spring predictability barrier for two types of El Niño events, Climate Dynamics, 47, 779–792, 2016.**

**Comment 2:** Lines 31-38: Using the NFSV-tendency assimilation approach, Duan and Tian (2013) revealed the dominant roles of zonal advection process in the development of CP-El Nino and the thermocline feedback process in the development of EP-El Nino events; then Duan et al. (2017) first demonstrated that the diversity of El Nino is closely related to changes in the nonlinear characteristics of the tropical Pacific.

- Duan W., B. Tian, and H. Xu. Simulations of two types of El Niño events by an optimal forcing vector approach.Climate Dynamics: 2013 ,DOI: 10.1007/s00382-013-1993-4
- Duan Wansuo, Chaoming Huang, Hui Xui, Nonlinearity modulating intensities and spatial structures of Central Pacific- and Eastern Pacific-El Niño events, Adv. Atmos. Sci., 2017. 34,737-756.

**Response:** Thank you for your introduction to these wonderful research work. According to your suggestion, we have supplemented the statements and cited these articles from the start of **line 40** as the **blue** text below:

"**Duan et al. (2013) proposed an optimal forcing vector (OFV) approach to optimize the Zebiak–Cane model and reproduced several observed EP and CP events, and revealed the dominant roles of zonal advection process in the development of CP-El Niño; then Duan et al. (2017) first demonstrated that the diversity of El Niño is closely related to changes in the nonlinear characteristics of the tropical Pacific.** *Accurate simulations and predictions of two types of ENSO are still of a great challenge, owing to the inherent uncertainty and diversity of ENSO (Chen and Cane, 2008; Trenberth and Stepaniak, 2001; Capotondi et al., 2015).*"

**References**

Duan, W., Tian, B., and Xu, H.: Simulations of two types of El Niño events by an optimal forcing vector approach, Climate dynamics, 43, 1677–1692, 2014.

Duan, W., Huang, C., and Xu, H.: Nonlinearity modulating intensities and spatial structures of central Pacific and eastern Pacific El Niño events, Advances in Atmospheric Sciences, 34, 737–756, 2017.

**Comment 3:** Line 32: GFDL abbreviation error.

**Response:** We are really sorry for our careless mistakes. Thank you for your reminder. We have reviewed the full manuscript and revised the abbreviation error at **line 36**.

**Comment 4:** Lines 51-52: Duan et al. (2004) is one of the earliest papers that explored the precursory disturbance of ENSO events (also see Duan et al., 2013)
- DuanW. , M. Mu, B. Conditional nonlinear optimal perturbation as the optimal precursors for El Nino-Southern Oscillation events. J. Geophy. Res.: 2004 ,109 ,D23105
- Duan W., Y. Yu, H. Xu, and P. Zhao. Behaviors of nonlinearities modulating El Nino events induced by optimal precursory disturbance. Climate Dynamics: 2013 ,40 ,1399–1413

**Response:** Thank you for your valuable suggestion. These references have been added and discussed at **line 59** as the blue text below:

"*In order to better understand the mechanism of ENSO occurrence, one approach is to explore the precursor of ENSO, which is the initial perturbation distribution that is most likely to develop into a CP event or an EP event.* **Duan et al. (2004) is one of the earliest papers that explored the precursory disturbance of ENSO events (Duan et al., 2013).** *These precursors help us understand the dynamic process of ENSO and provide the potential to predict ENSO events and their types.*"

**References**

Duan, W., Mu, M., and Wang, B.: Conditional nonlinear optimal perturbations as the optimal precursors for El Nino–Southern Oscillation events, Journal of Geophysical Research: Atmospheres, 109, 2004.

Duan,W., Yu, Y., Xu, H., and Zhao, P.: Behaviors of nonlinearities modulating the El Niño events induced by optimal precursory disturbances, Climate Dynamics, 40, 1399–1413, 2013.

**Comment 5:** Line 108: The descriptions of skip-layer connection structure and its attention mechanism are insufficient, and it is recommended to add implementation details.

**Response:** Thank you for the above suggestion. We have added a figure to elaborate the implementation details of skip-layer connection and the attention mechanism. For the encoder-decoder model architecture, skip-layer connection is used to transfer the multi-scale features extracted from the encoder layer to the decoder and help restore the fine spatial information. The time dimension of the original features obtained from the encoder is the input length of model input data $T_{in}$. In order to make the model automatically learn the influence of features of different lead months on prediction, we designed an attention mechanism to process the transferred features in the skip-layer connection. As shown in Fig. 12, a two-layer densely-connected neural network is used for original features to obtain the attention weight $\beta$. The original feature maps of each time step

are multiplied by the corresponding weights to get the weighted features for the decoder.

The detailed description and the related figure have been supplemented in the Section 3 from the start of **line 232** as the blue text below:

[Figure]

**Figure 12.** The detailed structure of the skip-layer connection and attention mechanism between encoder and decoder at the $n^{th}$ layer in ENSO-MC.

"The ENSO-MC model learns the feature of ENSO at different spatial scales with the convolution and pooling layers in the encoder, and gradually restores the spatial dimensionality of the original SST field in the decoder. With symmetrical structure design of the encoder and decoder as shown in Fig. 1, skip-layer connection is used to transfer features form the encoder to the decoder to recover spatial information lost during downsampling (yellow line in Fig. 1). Rather than transferring the original features of all time steps obtained from the encoder, we design an attention mechanism to enable the skip-layer to automatically learn the attention weights $\beta_1, \beta_2, ..., \beta_t$ on the temporal sequence because these air-sea features may have different effects on ENSO development at different time scales. As shown in Fig. 12, the encoder obtains the features $f_n \in \mathbb{R}^{T_{in} \times h_n \times w_n \times c_n}$ after maxpooling and convolution calculation at the $n^{th}$ layer. Using a two-layer densely-connected neural network, we obtain the attention weight $\beta \in \mathbb{R}^{T_{in}}$ of each time step's features according to Eq. (1), where $f'_n \in \mathbb{R}^{T_{in} \times (h_n \times w_n \times c_n)}$ are reshaped from $f_n$:

$$\beta = \text{softmax}\left(\mathbf{W}_{\beta\alpha} \tanh\left(\mathbf{W}_{\alpha f} f'_n + \mathbf{b}_{\alpha f}\right) + \mathbf{b}_{\beta\alpha}\right), \tag{1}$$

where $\mathbf{W}_{\alpha f}$, $\mathbf{W}_{\beta\alpha}$ are weight matrices created by the layer, and $\mathbf{b}_{\alpha f}$, $\mathbf{b}_{\beta\alpha}$ are the bias vectors. $\beta$ represents the contribution of each time step to prediction. According to Eq. 2, the feature maps of each time step are multiplied by the corresponding weights, and the fused maps $\tilde{f}_n \in \mathbb{R}^{h_n \times w_n \times c_n}$ are obtained by adding them along the time dimension.

$$\tilde{f}_n = \sum_{T_{in}} (\beta \circ f_n), \tag{2}$$

where $\tilde{f}_n$ are the feature maps to be transmitted in the skip-layer connection, which are connected to the features of the corresponding layer in the decoder. Besides, we also add states connection between the encoder and the decoder (grey line in Fig. 1), where the hidden states output by the ConvLSTM layers in the encoder are reserved for the corresponding layer when the decoder is initialized. With the methods of skip-layer connection and states connection, the model can make

full use of the information extracted from the encoder before ENSO events, which help stabilize training and convergence."

**Comment 6:** Line 140: Specific explanation should be given for the meaning represented by SSIM, and clarify why it should be used as a loss function in ENSO prediction.

**Response:** Thank you very much for your kindly suggestions. We agree that we have neglected to elaborate on the meaning of the loss functions and the impact of their choices on the forecasts. SSIM loss function is used to reduce the global structural differences between prediction and observation fields. The main indicators are luminance, contrast and structure, namely the mean value and standard deviation of a field, and the covariance of the two fields. Since ENSO is closely related to the region of equatorial Pacific where the maximum variance of SST is located, and the mean of SST anomalies in some region is commonly used as an indicator of event occurrence, we use the loss function based on SSIM to measure the global difference.

We have added the specific meaning of original SSIM metric and its implications for ENSO prediction, and discussed the effects of different loss functions on our training results. The detailed description has been supplemented in the Section 2 from the start of **line 177** as the blue text below:

"In addition to quantifying difference in each corresponding pixel value between the observations and predictions, we introduce a loss based on SSIM to measure the global structural differences. SSIM is widely used as a metric to measure the similarity of two images by extracting structural information. It takes into account three features: luminance (l), contrast (c) and structure (s), and its metric formula is the product of these three elements.

$$SSIM(x,y) = \left(\frac{2\mu_x\mu_y+C_1}{\mu_x^2+\mu_y^2+C_1}\right)_l \cdot \left(\frac{2\sigma_x\sigma_y+C_2}{\sigma_x^2+\sigma_y^2+C_2}\right)_c \cdot \left(\frac{\sigma_{xy}+C_3}{\sigma_x\sigma_y+C_3}\right)_s = \frac{(2\mu_x\mu_y+C_1)(2\sigma_{xy}+C_2)}{(\mu_x^2+\mu_y^2+C_1)(\sigma_x^2+\sigma_y^2+C_2)}, \tag{5}$$

where $\mu$ is the mean value of a field (luminance), $\sigma$ is the standard deviation (contrast) and $\sigma_{xy}$ is the covariance of the two fields (structure). $C_1$, $C_2$, $C_3$ are constants used to maintain the calculations stable. ENSO is associated with the interannual variations of SST anomalies in the tropical Pacific. And the Nino3.4 index is one of the most commonly used ENSO indicators, which is the average SST anomalies in the equatorial Pacific sub-region where the maximum variance of SST is located. Therefore, SSIM metric can help evaluate important signals embedded in the SST patterns for ENSO prediction (Mo el al., 2014). The range of SSIM is from 0 to 1, and when two fields are the same, the value of SSIM is 1. Therefore, we construct the SSIM-based loss function as

$$\mathcal{L}_{ssim}(Y,\hat{Y}) = \frac{1-\frac{1}{T}(\sum_{t=1}^{T}SSIM(Y_t,\hat{Y}_t))}{2}. \tag{6}$$"

**References**

Mo, R., Ye, C., and Whitfield, P. H.: Application potential of four nontraditional similarity metrics in hydrometeorology, Journal of Hydrometeorology, 15, 1862–1880, 2014.

The effects of different loss functions on the training results are also additional supplemented at the end of Section 3 at the **line 258** as the blue text below:

[Figure]

**Figure 13.** The performances of the ENSO-MC with different loss functions.

"We validate the effectiveness of combined loss function. As shown in Fig. 13(a), although SSIM and GDL do not significantly improve the model performance when combined with MSE alone, the combination of MSE, SSIM and GDL loss functions achieve the best performance on the correlation skill. Besides, since GDL loss function tends to retain extreme values and MSE loss function tends to smooth all values, the presence of GDL inhibits the decrease of MSE, so the MSE errors of the models with GDL loss function are higher than the ones without GDL (Fig. 13(b)). And comparing the results of correlation skill and RMSE in Fig. 13(a) and (b), low RMSE values do not represent high correlation skills. Therefore, it is necessary to explore loss functions suitable for ENSO prediction other than MSE to balance the training of the model."

**Comment 7:** Line 147: For "gradient information of gridded variables is important for the model to understand changes in the sea temperature", how does this conclusion come? Please explain why gradient information is necessary.

**Response:** Thank you for pointing out this problem in manuscript. We are very sorry for our ambiguous statements of GDL loss. The SST gradient represents the difference in the sea temperature across the adjacent area. MSE loss function tends to average the values of all points in the whole prediction field to minimize the MSE error (Opera et al., 2020), which is not conductive to the prediction of ENSO extreme values, while considering the gradient difference value can alleviate this problem. Besides, the SST gradient also plays a role in the atmospheric circulation. The region with a large SST gradient will generate stronger winds, which in turn promote the further increase of the SST gradient (Bjerknes, 1969). As ENSO approaches maturity, the SST gradient increases gradually. Here we only consider the gradient difference in neighboring regions. In future studies, we will consider the gradient difference at a larger spatial scale according to the characteristics of ENSO, such as the difference in SST between the Western Pacific and the Central Pacific during ENSO (Zinke et al., 2021).

Following your suggestion, we have supplemented a more detailed description of GDL loss and the related results of training performance in the Section 2 from the start of **line 192** as the blue text below:

"We also consider gradient information in the loss functions. The SST gradient represents the

difference in the sea temperature across the adjacent area. Previous studies have shown that MSE loss function tends to average the values of all points in the whole prediction field to minimize the MSE error, while considering the gradient difference value can alleviate this problem (Opera et al., 2020). Besides, the SST gradient also plays a role in the atmospheric circulation. The region with a large SST gradient will generate stronger winds, which in turn promote the further increase of the SST gradient (Bjerknes, 1969). As ENSO approaches maturity, the SST gradient increases gradually. Therefore, we use the GDL to measure the gradient difference of the surface sea temperature field:

$$\mathcal{L}_{gdl}(Y, \hat{Y}) = \frac{1}{T}\sum_{t=1}^{T}\sum_{i,j}\left\||Y_{i,j}^{t} - Y_{i-1,j}^{t}| - |\hat{Y}_{i,j}^{t} - \hat{Y}_{i-1,j}^{t}|\right\|^{2} + \left\||Y_{i,j-1}^{t} - Y_{i,j}^{t}| - |\hat{Y}_{i,j-1}^{t} - \hat{Y}_{i,j}^{t}|\right\|^{2} \quad (7)$$

where $i, j$ denote the pixel position on the sea surface temperature field. Here we only consider the gradient difference in neighboring regions. In future studies, we will consider the gradient difference at a larger spatial scale according to the characteristics of ENSO, such as the difference in SST between the Western Pacific and the Central Pacific during ENSO (Zinke et al., 2021).

[Figure]

**Figure 13.** The performances of the ENSO-MC with different loss functions.

We validate the effectiveness of combined loss function. As shown in Fig. 13(a), although SSIM and GDL do not significantly improve the model performance when combined with MSE alone, the combination of MSE, SSIM and GDL loss functions achieve the best performance on the correlation skill. Besides, since GDL loss function tends to retain extreme values and MSE loss function tends to smooth all values, the presence of GDL inhibits the decrease of MSE, so the MSE errors of the models with GDL loss function are higher than the ones without GDL (Fig. 13(b)). And comparing the results of correlation skill and RMSE in Fig. 13(a) and (b), low RMSE values do not represent high correlation skills. Therefore, it is necessary to explore loss functions suitable for ENSO prediction other than MSE to balance the training of the model."

**References**

Oprea, S., Martinez-Gonzalez, P., Garcia-Garcia, A., Castro-Vargas, J. A., Orts-Escolano, S., Garcia-Rodriguez, J., and Argyros, A.: A review on deep learning techniques for video prediction, IEEE Transactions on Pattern Analysis and Machine Intelligence, 2020.

Bjerknes, J.: Atmospheric teleconnections from the equatorial Pacific, Monthly weather review, 97, 163–172, 1969.

Zinke, J., Browning, S., Hoell, A., and Goodwin, I.: The West Pacific Gradient tracks ENSO and zonal Pacific sea surface temperature gradient during the last Millennium, Scientific reports, 11, 1–16, 2021.

**Comment 8:** Line 225: It is suggested to clarify whether the precursor analysis is based on a multi-step forecast strategy model or a one-step model.

**Response:** Thank you for spotting our crucial neglects in precursor analysis. Since the ENSO-MC model performs better using the multi-step forecast strategy, we calculate the saliency maps for precursor analysis and sensitive area identification based on the multi-step forecasting model. The related statements have been supplemented at the **line 370** as the **blue** text below:

"*Based on the ENSO-MC model that successfully simulates different types of ENSO events, we can further explore the ENSO dynamics learned by the ENSO-MC model and observe the signals before the onset of events.* **And since the ENSO-MC model using the multi-step forecast strategy achieves better performance than using one-step strategy, here we calculate the saliency maps based on the multi-step forecasting model for precursor analysis and sensitive area identification.**"

**Comment 9:** Line 273-274: Here the first two areas with the highest sensitivity are selected as sensitive areas. If the first three or more are selected, will the benefit be higher?

**Response:** Thanks for bringing up this issue that we didn't illustrate in the previous manuscript. Fig. 20(b) shows the sensitivity of six given regions we calculated through sensitivity experiments.

[Figure]

**Figure 14.** Sensitive areas identification results for ENSO with the saliency map method.

In our manuscript, the first two areas with the highest sensitivity of surface as well as subsurface are selected as the sensitive areas of targeted observation for ENSO. Due to the diversity of sensitive areas for ENSO, and the fact that we want to explore the distribution of sensitive areas in the tropical Pacific as well as the subtropical Pacific, the number of sensitive areas we identify is greater than 1. Following your suggestion, we select the first three most sensitive areas to evaluate the benefit of the identification, that is, the area_0, area_1 and area_2 for surface and the area_1, area_3 and area_4 for subsurface as shown in Fig. 14(b).

[Figure]

**Figure 15.** The benefit for removing the random perturbations in the sensitive areas (blue) and removing the random perturbations outside the sensitive areas (orange) for the eight El Niño events and three La Niña events. (a) uses the first two areas with the highest sensitivity. (b) uses the first three areas.

The results are illustrated in Fig. 15(b). Compared with the benefit of two sensitive areas (Fig. 14(a)), the benefits for most ENSO events do not increase significantly after adding one more sensitive area, except that the events in 2002 and 2010 each increase by about 7%. Therefore, we think it is appropriate and reasonable to choose the first two areas here.

**Comment 10:** Figure 5: How can we know that the forecasting skills of this model decline fastest in the late boreal spring? please provide a clear analysis. In Fig. 5(d), the forecasting skills improve slightly after 12 months in the one-step strategy model. Why does this happen?

**Response:** Thank you for pointing out this problem in manuscript. We have marked the original Fig. 5 and supplemented Fig. 16 to present the decline of forecasting skills in each target month compared with the previous month in multi-step time series forecasting. The prediction skills of the model decline most from April to May, regardless of whether the multi-step or one-step ahead forecast strategy is used. It makes the forecasting skills of the model for ENSO are reduced to the lowest in May and June, as marked by the black numbers in Fig. 5. Besides, Fig. 16 shows that the performance of the model is slightly improved in winter, which leads to the improvement of skills after 12 months in Fig. 5. Since the seasonal variation of SST anomaly variance is weaker in spring, it is difficult for the model to capture useful information, which leads to the spring predictability barrier, while the strong signals of ENSO during winter are more easily learned by the model.

The analysis and the related figures have been modified and supplemented from the start of **line 350** as the blue text below:

[Figure]

**Figure 5.** The correlation skills of the Nino3.4 index forecasts started from each calendar month in ENSO-MC using multi-step forecast strategy (a) and one-step ahead forecast strategy (b) for the GODAS data from 1982 to 2019. (c) is the same as (a), except for the GODAS data from 2010 to 2019. (b) is the same as (d), except for the GODAS data from 2010 to 2019. Hatches represent the forecasts with correlation skill exceeding 0.5, and the black numbers mean the target forecast months.

[Figure]

**Figure 16.** The decline of forecasting skills for ENSO in each target month using multi-step forecast strategy and one-step ahead forecast strategy.

"*Figures 5(a) and 5(c) are the results of multi-step prediction, while Figures 5(b) (1982-2019) and 5(d) (2010-2019) are the results of one-step prediction.* It shows that regardless of the season from which the forecast is started, the skills would be reduced for predictions targeting the late boreal spring (April-May–June, AMJ), as indicated by the black numbers in Fig. 5. We also calculate the overall decline of forecasting skills in each target month compared with the previous month and the results are presented in Fig. 16. The prediction skills of the model decline most from April to May, regardless of whether the multi-step or one-step ahead forecast strategy is used. Besides, Fig. 16 shows that the performance of the model is slightly improved in winter, which leads to the improvement of skills after 12 months in Fig. 5. Since the seasonal variation of SST anomaly

variance is weaker in spring, it is difficult for the model to capture useful information, which leads to the spring predictability barrier (SPB), while the strong signals of ENSO during winter are more easily learned by the model. In addition, the one-step ahead strategy has a larger decline after the boreal spring (Fig. 16), and the subsequent forecasts are more susceptible to the SPB due to its cumulative error (Figs. 5(b), (d)), *while the method of the multi-step strategy can reduce the influence (Figs. 5(a), (c)). We can conclude that the ENSO-MC model using multi-step forecast strategy is less affected by spring predictability barriers.*

**Comment 11:** Figure 7, 8: It is suggested to further clarify whether the saliency value in the figure is the result after standardization or the original perturbation amplitude of SST and heat content.

**Response:** Thank you so much for your professional attitude and spotting our neglects in figures. The saliency values in the figures are standardization results of scaling between 0 and 1. We have supplemented the related statements at **line 453** as the **blue** text below:

"*Then all the saliency maps of SST are added up to obtain the composite saliency map of the surface (Fig. 7(a)), and that of the subsurface (Fig. 7(b)) is obtained in the same way.* **The saliency values in the figure are the standardized results of the scale between 0 and 1.**"

Thank you again for your positive comments and valuable suggestions to improve the quality of our manuscript.

On behalf of all the co-authors, best regards,
Yuehan Cui